

# Brown bear communication hubs: patterns and correlates of tree rubbing and pedal marking at a long-term marking site

Eloy Revilla[1], Damián Ramos Fernández[2], Alberto Fernández-Gil[1], Agnieszka Sergiel[3], Nuria Selva[3] and Javier Naves[1]

[1] Department of Conservation Biology, Estación Biológica de Doñana CSIC, Seville, Spain
[2] Consejería de Infraestructuras, Ordenación del Territorio y Medio Ambiente, Gobierno del Principado de Asturias, Oviedo, Spain
[3] Institute of Nature Conservation, Polish Academy of Sciences, Krakow, Poland

Corresponding author
Eloy Revilla, revilla@ebd.csic.es

## ABSTRACT

Chemical communication is important for many species of mammals. Male brown bears, *Ursus arctos*, mark trees with a secretion from glands located on their back. The recent discovery of pedal glands and pedal-marking at a site used for tree-rubbing led us to hypothesize that both types of marking form part of a more complex communication system. We describe the patterns of chemical communication used by different age and sex classes, including differences in the roles of these classes as information providers or receivers over four years at a long-term marking site. Using video recordings from a camera trap, we registered a total of 285 bear-visits and 419 behavioral events associated with chemical communication. Bears visited the site more frequently during the mating season, during which communication behaviors were more frequent. A typical visit by male bears consisted of sniffing the depressions where animals pedal mark, performing pedal-marking, sniffing the tree, and, finally, rubbing against the trunk of the tree. Adult males performed most pedal- and tree-marking (95% and 66% of the cases, respectively). Males pedal-marked and tree-rubbed in 81% and 48% of their visits and sniffed the pedal marks and the tree in 23% and 59% of visits, respectively. Adult females never pedal marked, and juveniles did so at very low frequencies. Females rubbed against the tree in just 9% of their visits; they sniffed the tree and the pedal marks in 51% and 21% of their visits, respectively. All sex and age classes performed pedal- and tree-sniffing. There were significant associations between behaviors indicating that different behaviors tended to occur during the same visit and were more likely if another individual had recently visited. These associations leading to repeated marking of the site can promote the establishment of long-term marking sites. Marking sites defined by trees and the trails leading to them seem to act as communication hubs that brown bears use to share and obtain important information at population level.

## INTRODUCTION

Marking behavior is essential in the mediation of chemical communication and social interactions in mammals (*Johansson & Jones, 2007*). The chemical signals left at specific sites provide long-lasting messages in the absence of the signal provider (*White, Swaisgood & Zhang, 2002*; *Scordato, Dubay & Drea, 2007*). In carnivores, the function of scent marks has been associated with territorial defense (*Wronski et al., 2006*), intra-sexual competition (*Gosling & Roberts, 2001*), and the defense of trophic resources (*Pineiro & Barja, 2015*). Scent marking is particularly important for solitary species ranging widely in large home ranges (*Begg et al., 2003*; *Vogt et al., 2014*). These species must rely on an effective communication system that maximizes the transfer of information at low cost in order to maintain their social organization by advertising to mates and competitors (*Allen, Yovovich & Wilmers, 2016*).

Urine and feces are a relatively inexpensive means of scent marking used by many carnivore species at the expense of relatively low efficiency in the transfer of information (*Vogt et al., 2016*). More specialized chemical compounds may provide detailed information on the individual, including their sex and reproductive status (*Alberts, 1992*). They are produced by specialized holocrine, apocrine and/or eccrine skin glands, often located in the anal, subcaudal, interdigital skin, and chin areas, among others. To be effective, their secretions should persist in the environment for long periods to maximize the probability of reaching potential receivers (*Swaisgood et al., 2004*). Additionally, individuals scent mark specific sites, such as territorial borders, and prominent locations that are often revisited by them and other individuals, including dens, food sources and busy trails (*Sillero-Zubiri & Macdonald, 1998*; *Revilla & Palomares, 2002*; *King et al., 2017*). Chemical cues guide receiving individuals to investigate, ignore, counter and/or over-mark previous marks (*Laidre & Johnstone, 2013*). The presence of long-lasting marks of multiple individuals in a marking area may promote the synergy between different types of signals, potentially eliciting several communication-related behaviors (*Sumpter & Brännström, 2008*). These complexities make some particular types of marking sites especially important in the regulation of social behavior. The repeated use by multiple individuals for long periods of time convert these marking sites into communication hubs at a population level (*King et al., 2017*).

Ursids are non-territorial animals that move over large areas with low contact rates between individuals (*Martin et al., 2013*). In spite of this, they maintain a complex network of social interactions in which information on the presence of other individuals is critical (*Støen et al., 2005*; *Steyaert et al., 2012*). Chemical communication plays an important role in the maintenance of bear social organization (*Noyce & Garshelis, 2014*). Brown bears *Ursus arctos* mark conspicuous objects such as trees, rocks or even poles, with secretions from the sebaceous glands and possibly also the apocrine glands located in the skin of their back (*Tomiyasu et al., 2018*), and, in some cases, with claw and bite marks as well (*Nie et al., 2012*; *Clapham et al., 2013*; *Taylor, Allen & Gunther, 2015*).

Bipedal back-rubbing against trees has been widely described as the most common marking behavior of brown bears across its Holarctic range, showing seasonal and sex

and age variations in marking frequency (*Green & Mattson, 2003*; *Clapham et al., 2012*; *Clapham et al., 2013*; *Sato et al., 2014*; *Seryodkin, 2014*; *Spassov et al., 2015*; *Tattoni et al., 2015*). Additionally, pedal-marking has recently been reported as an important marking behavior (*Taylor, Allen & Gunther, 2015*; *Sergiel et al., 2017*). Typical deep marks left in the ground by brown bears, possibly during pedal-marking, were described long ago as leading towards bear trees (*LeFranc Jr et al., 1987*). The presence of pedal scent glands in brown bears and their significance in communication have also been recently described (*Sergiel et al., 2017*). Nevertheless, pedal-marking has yet to be characterized in terms of its phenology, the sex and age class of the individuals and other environmental correlates, as well as its connection with tree marking, given that they seem to simultaneously occur at the same sites (*Clapham et al., 2014*; *Sergiel et al., 2017*).

In this paper we hypothesize that pedal-marking and tree-rubbing are deeply linked in brown bears, forming a more complex communication system than previously recognized. We expect to find differences in the use of marking sites by different sex and age classes of individuals, depending on their primary role as either information providers or receivers. Specifically, we made use of a multi-year dataset on chemical communication by brown bears at a marking site in a well-known population living in the Cantabrian Mountains, northern Spain. The site is known to have been intensively used for pedal-marking and tree-rubbing by bears since 2002, when it was already well established, and has therefore been used by more than a generation (see *Sergiel et al., 2017* for a basic description of pedal marking at this site). Specifically, we aimed at (1) assessing the frequency of main marking behaviors by brown bears of different age and sex classes; (2) identifying associations among behaviors as well as among signal providers (the ones marking) and receivers (the ones sniffing the marks), and (3) determine the role of other factors, such as climatic variables, in the occurrence of marking behaviors. Finally, we discuss the significance of these communication hubs intensively used by brown bears for long periods of time.

## MATERIALS & METHODS

### Study site
The study was conducted in the western half of the Cantabrian Range (NW Spain), a mountain system inhabited by a brown bear population which currently numbers around 230 individuals, with a density of 1.6 individuals/100 km$^2$ (*Pérez et al., 2014*). The study area is located in Fuentes del Narcea, Degaña e Ibias Natural Park (Cangas del Narcea, Asturias). Our study site is located in an area with high quality habitat for brown bears (*Naves et al., 2003*), including denning and mating areas, areas used by females with cubs, and also vegetation offering plenty of resources used during hyperphagia, when bears feed continuously in preparation for hibernation.

In this area, there are multiple sites used by brown bears for chemical communication. These sites can be easily identified by the presence of a tree, pole or rock that is used for rubbing, often in association with a series of pedal marking tracks leading to the vertical structure that is marked. We selected one site for continuous monitoring on the basis of the evidence of repeated use by brown bears for pedal-marking for more than a decade

(*Sergiel et al., 2017*). As the Cantabrian brown bear population is threatened, we do not provide the exact location of the site due to conservation concerns. The first evidence of ground pedal-marking at this site was obtained in 2002 during an opportunistic observation by one of the authors (DR) of an adult male during the mating season. The site is characterized by an oak tree (*Quercus petraea*) heavily used by brown bears for rubbing, and by conspicuous marks in the ground made by the bears' repeated use of the same spots for pedal-marking (a total of 48 marks made by bears' feet are evident to the human eye).

## Sampling protocol

Data were collected by DR at the selected site during long-term monitoring for conservation and management. The Principado de Asturias–Consejería de Agroganadería y Recursos Autóctonos granted data access, and DR was authorised to participate by exp-no. 2016/033072, Principado de Asturias-Consejería de Hacienda y sector Público. An automatic camera trap (Bushnell Trophy digital camera trap #19466 with motion triggered day/night recording) was set up between January 2012 and January 2016, during which time it was working almost continuously. Initially, between January 2012 and April 2012, the device was placed laterally in a low position from which the tree marked by brown bears was visible. Data obtained during these first four months were not used in the analyses. After this initial sampling, the camera trap was mounted in a zenith position (directly above the site) at a height of six meters on the main trunk of the marked tree to obtain a standardized field of view and to reduce direct interference with bears and other animals. The field of view of the camera trap covered an area of about 100 m$^2$. The camera trap was programmed to shoot one-minute videos, with a 10-second interval between consecutive videos. We considered a visit event as the group of videos that are not more than 20 min apart. This time window was selected following visual inspection of the plot of the cumulative proportion of videos sorted by the time to the next video (Fig. S1). For comparative purposes, we also used a 20-minute time interval to define visit events for other species. Note that a visit can include more than one individual bear, as occurs in the case of females with cubs or males and females moving together during the mating season. The weather data were obtained from the nearby automatic station of Leitariegos, belonging to the Spanish Agencia Estatal de Meteorología (AEMET).

## Individuals and communication behaviors

In the Cantabrian Mountains, the steep slopes and low forest cover make it relatively easy to observe brown bears, especially during spring and summer. Individuals present in valleys are detected by scanning the area with spotting scopes from vantage points. This method is used to obtain annual counts of the number of females with cubs of the year and as a long-term method to census this population (*Wiegand et al., 1998*). As a result, some of the individuals moving in the study area are known, especially when they have some identifying marks, and are thus easily distinguished from other individuals. The professional technicians doing those censuses are experts in recognizing the sex and age of individuals by specific traits under good observation conditions. We classified the recorded individuals into the following sex and age categories: (1) adult males, identified by the

combination of large size, and neck and head shape; (2) adult females, when accompanied by cubs, or identified by their size, head and neck shapes, and explicit behavior in the presence of other individuals, often adult males in the mating season; (3) cubs, individuals in their first year or in their second year until May and always accompanied by their mother; (4) juveniles, independent individuals in their second year of life from June onwards and in their third year, clearly smaller in size than adults and usually accompanied by siblings; and, (5) undetermined sex and age class, which included the remaining individuals.

In the case of adult males, some individuals were identified by comparison with known animals observed in repeated sightings at other sites in the study area. These individuals were characterized by a combination of body size, head shape, coat color patterns and especially the very characteristic light-colored permanent markings, normally present on their necks (see description of individualized bears in Supplemental Information). The Cantabrian brown bear population is characterized by its small size and the large variability shown by individuals in coat color and the common presence of markings especially on their necks (*Clevenger & Purroy, 1991*). In other cases, we were able to temporarily classify some individuals in an age and sex class or even identify them during shorter periods of time because they were associated with other bears in seasonal or yearly groups such as mating pairs, females with dependent cubs, and groups of independent juveniles repeatedly seen in the area. Females are more difficult to identify individually on a permanent basis. We used the number of accompanying cubs to establish a minimum number of females visiting the site each year. We did not attempt to identify other types of individuals such as independent juveniles and cubs.

We classified the behaviors displayed by brown bears in the videos into the following types: (1) sniffing pedal marks, when an individual stops or slows its pace and puts its nose to the pedal marks on the ground; (2) pedal-marking, performed by a walking bear with the particular gait of twisting its fore and hind feet on the ground in specific depressions repeatedly used by that individual and other bears during previous visits; (3) tree-sniffing, when an individual calmly puts its nose to the trunk of the rubbing tree; (4) tree-rubbing, when a bear vigorously rubs its back, neck or shoulders against the trunk of the tree while standing on its hind legs; and, (5) other behaviors, in which a bear usually walks in and out of the field of vision. In the videos recorded at the study site we did not detect any clear instance of scratching the tree (clawing; *Taylor, Allen & Gunther, 2015*). For each visit event we determined if each type of behavior was performed (presence/absence of the behavior, not the number of times) by each bear in the available sequence of videos.

## Analyses

First, we described the overall use of the site and the behaviors performed by the visiting brown bears over time and by age and sex classes. Then, we analyzed which variables were associated with the observed patterns (Table 1). We hypothesized that the probability that bears visited the marking site in a given day and performed one of the behaviors in each visit was affected by not only the time elapsed since the previous visit by a bear, but also the season, distinguishing between mating season (April, May and June) and non-mating season (other months), as well as the age and sex class of the focal individuals, and, in the

**Table 1 Description of response and explanatory variables used in the analyses.** All response variables were binary: occurrence of visit or visits in a given day for *day visit* or occurrence within a visit for communication behaviors. The variables listed were the ones explored in each model (marked with $X$). Not all combinations were explored due to biological sense (weather variables were used only for sniffing behaviors because weather can affect the amount of time that marks last, or due to the most common logical sequence of events, from sniff pedal marks into tree rubbing), or to the structure of the data (*day visit* has no individual descriptors as in a given day more than one individual can occur; *pedal marking* can only be analysed for males because they were the only ones using this marking).

| | Explanatory variables | | Response variables | | | | |
|---|---|---|---|---|---|---|---|
| **Label** | **Description** | | *Day visit* | *Sniff pedal marks*[b] | *Pedal marking*[a] | *Sniff tree*[b] | *Tree rubbing*[b] |
| Individual variables | | | | | | | |
| *age_sex* | Age-sex class of the bear (Male, Female, Juvenile, Undetermined) | | | X | | X | X |
| *age_sex_tree* | Age-sex class of the previous bear marking the tree (Male, Female, Juvenile, Undetermined) | | | | | X | |
| Temporal variables | | | | | | | |
| *days* | Time since the previous visit of a bear (in days, common logarithm) | | | X | X | X | X |
| *days_male* | Time since the previous visit of a male (in days, common logarithm) | | X | | | | |
| *days_pedal* | Time since the previous visit of a bear pedal marking (in days, common logarithm) | | X | X | X | | |
| *days_tree* | Time since the previous visit of a bear rubbing the tree (in days, common logarithm) | | X | | | X | X |
| Weather variables | | | | | | | |
| *Prec_pedal* | Average precipitation of the days elapsed since the previous bear visit that performed pedal marking (mm) | | | X | | | |
| *Prec_tree* | Average precipitation of days elapsed since the previous bear visit that performed tree marking (mm) | | | | | X | |
| *Temp_pedal* | Average temperature of the days elapsed since the previous bear visit that performed pedal marking (°C) | | | X | | | |
| *Temp_tree* | Average temperature of the days elapsed since the previous bear visit that performed tree rubbing (°C) | | | | | X | |
| Behavioral variables | | | | | | | |
| *pedal_marking* | Pedal marking performed by the same bear visit | | | | | | X |
| *season* | Season: mating (April, May, June) vs non-mating (other months) | | X | X | X | X | X |
| *sniff_pedal* | Sniff pedal marks during the bear visit | | | | X | | |
| *sniff_tree* | Sniff tree during the bear visit | | | | | | X |
| *tree_rubbing* | Tree-rubbing during the same bear visit | | | | X | | |

**Notes.**
  [a]Only for males.
  [b]All bears except cubs.

analyses where it made biological sense, by the weather conditions that occurred between visit events affecting the duration of the chemical signals. We performed Generalized Linear Mixed Models (GLMMs) on the response variables (occurrence of the specific behaviors) using a binomial error distribution and year as a random factor. As some individuals were repeatedly observed, there could be some pseudoreplication problem. Solving this issue is not easy as a fraction of the observations correspond to unknown
animals. Nevertheless, and in order to check if pseudoreplication was an issue, we repeated the analyses of the selected models adding individual ID as an additional random factor (unidentified individuals were grouped under a single individual label; results are shown in the Supplemental Information). Models were run with the potential combination of biologically meaningful explanatory variables within each group of response variables (Table 1). To reduce the effect of multicolinearity, when two predictors were correlated, we selected the one with a stronger association with the dependent variable. From the resulting models, we report and interpret only those within $\Delta$AIC < 2. We computed the marginal and conditional $R^2$ for each selected model (*Nakagawa & Schielzeth, 2013*). For inference we used a cut-off level of $P < 0.10$. Analyses were performed in *R* vs3.3.3 with lme4 (v1.1-19; *Bates et al., 2015*) and MuMIn (v1.43.6; *Bartoń, 2019*) packages.

# RESULTS

In total, the camera trap was active for 1,174 days (April 2012 to December 2015), with an average temporal coverage of 83% of the possible days per month (Fig. 1, Table S1). It registered 329 videos with bear presence; representing 224 visit events and a total of 285 bear-visit events (note that more than one individual can be present during the same visit event). Brown bears were the most common visitors (42%), with more than five visits per month on average (Fig. S2, Table S2). The visitation rate of other species was considerably lower despite being more abundant in most cases (Fig. S2). Among brown bears, adult males were the most frequent visitors with 132 bear visits (46% of total bear visits, Table 2). The rest of the visits were performed by adult females in 57 cases (20%), cubs in 44 (15%), juveniles in 23 (8%) and individuals of undetermined age and sex in 29 (10%; Table 2). The visits followed a bimodal diel pattern with maxima around dawn and dusk (*Kaczensky et al., 2006*) but occurred also throughout the day and into the night (24-h rhythm, Fig. S3). Bears visited the marking site more frequently during the mating season (April–June; Fig. 1, on average 26.3% of the days sampled per month had bears visiting the site during the mating season, versus 14.1% during the rest of the year excluding the hibernation period (January and February); Table 3; Table S3). The probability that the site was visited by bears on a given day was negatively associated with the time since the last visit of a male (the shorter the lapse, the higher the probability; Table 3; Table S3).

## Communication behaviors

The typical sequence of a visit consists of a bear approaching the tree following the path where it can sniff the depressions in which animals pedal mark, performing pedal-marking itself, stopping at the tree, sniffing it, and, finally, rubbing against the trunk (see Video S1). This sequence can vary with different combinations of behaviors and in different orders, and some parts of the sequence can be repeated. On one occasion, a male also rubbed its body against pedal marks. There was no apparent communication behavior in 22% of the visits, although some could have occurred out of the field of view of the camera trap.

Out of a total of 482 recorded behaviors, the majority corresponded with some form of chemical communication (87%). Communication behaviors occurred in most months except January and February (hibernation period; Table S4). Sniffing of pedal marks was

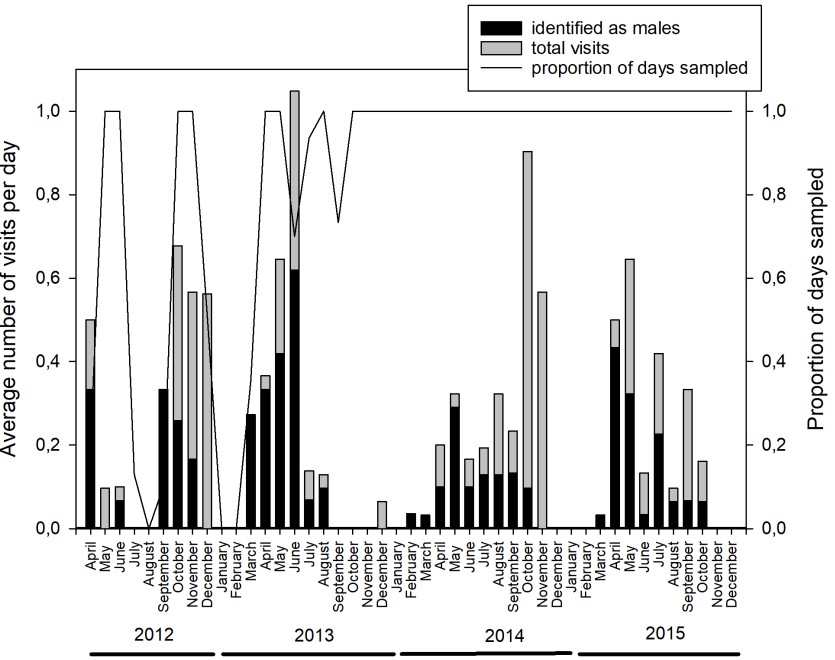

**Figure 1** **Monthly distribution of brown bear visits to the marking site.** The average number of individual visits per day of sampling (left axis, indicating the total number and those identified as males) and the sampling effort (right axis), measured as the proportion of days that the camera trap was active every month (*x*-axis) between April 2012 and December 2015. See Table S1 for numerical data.

**Table 2** **Number of behaviours displayed by different age and sex classes.** Data recorded by the camera trap at the marking site between 2012 and 2015.

| Behavior | Age-Sex classes | | | | | Total |
|---|---|---|---|---|---|---|
| | **Males** | **Females** | **Cubs** | **Juveniles** | **Undetermined** | |
| Sniffing pedal marks | 30 | 12 | 3 | 9 | 4 | 58 |
| Pedal-marking | 107 | 0 | 0 | 4 | 2 | 113 |
| Sniffing tree | 78 | 29 | 27 | 11 | 8 | 153 |
| Tree-rubbing | 63 | 5 | 15 | 9 | 4 | 96 |
| Other | 5 | 20 | 12 | 7 | 18 | 62 |
| Total number of behaviors | 283 | 66 | 57 | 40 | 36 | 482 |
| Total number of visits | 132 | 57 | 44 | 23 | 29 | 285 |

less frequent (58, 12%) than pedal-marking (113, 23%); while tree-sniffing (153 cases, 31%) was more frequent than tree-rubbing (96, 20%; Table S4).

The communication behaviors displayed by bears varied greatly among age and sex classes. All sex and age classes performed pedal- and tree-sniffing. Individuals identified as adult males performed most of the pedal-marking (107 cases, 95%) and, to a lesser extent, tree-rubbing (63 cases, 66%, Fig. 2). Interestingly, adult females did not perform pedal-marking, while juveniles did so at very low frequency (Fig. 2). Tree-rubbing was performed by all age and sex classes, but at higher frequencies by males (Fig. 2).

**Table 3 Estimates of the effect of the factors included in the best models.** Models were GLMMs with binomial distribution and year as random factor (Table S7). The models on pedal marking were run only on males and the rest with all types of individuals except for cubs. Note that, for the *tree rubbing* model, the estimates of the *age_sex* parameters correspond with the comparison of those classes with females, which is the reference class (as defined by the intercept of the model). See Table 1 for a description of the variables.

| Model | Estimate | SE | p |
|---|---|---|---|
| *Bear visit* (all classes of individuals) | | | |
| (Intercept) | 0.859 | 0.267 | 0.001 |
| days_male | −1.823 | 0.196 | <0.0001 |
| Season | −0.379 | 0.177 | 0.032 |
| $R^2$ (marginal) = 0.30 | | | |
| $R^2$ (conditional) = 0.30 | | | |
| *Sniff pedal marks* (all classes of individuals except cubs) | | | |
| (intercept) | −2.069 | 0.797 | 0.009 |
| days_pedal | −0.725 | 0.389 | 0.062 |
| Prec_pedal | −0.013 | 0.006 | 0.036 |
| Temp_pedal | −0.011 | 0.005 | 0.013 |
| Season | 2.046 | 0.546 | <0.001 |
| $R^2$ (marginal) = 0.21 | | | |
| $R^2$ (conditional) = 0.26 | | | |
| *Pedal marking* (males) | | | |
| (Intercept) | 1.946 | 0.494 | <0.0001 |
| days_pedal | −1.255 | 0.477 | 0.009 |
| tree_rubbing | 1.315 | 0.527 | 0.013 |
| $R^2$ (marginal) = 0.20 | | | |
| $R^2$ (conditional) = 0.20 | | | |
| *Sniff tree* (all classes of individuals except cubs) | | | |
| (Intercept) | −0.090 | 0.249 | 0.717 |
| Days | 0.885 | 0.379 | 0.019 |
| Prec_tree | −0.011 | 0.005 | 0.047 |
| $R^2$ (marginal) = 0.06 | | | |
| $R^2$ (conditional) = 0.06 | | | |
| *Tree rubbing* (all classes of individuals except cubs) | | | |
| (Intercept) | −3.611 | 0.651 | <0.0001 |
| days_tree | 0.857 | 0.461 | 0.063 |
| sniff_tree | 1.412 | 0.352 | <0.0001 |
| pedal_marking | 1.293 | 0.502 | 0.010 |
| age_sex | | | |
| Undetermined | 0.378 | 0.771 | 0.624 |
| Juvenile | 1.790 | 0.753 | 0.018 |
| Male | 1.146 | 0.666 | 0.086 |
| $R^2$ (marginal) = 0.36 | | | |
| $R^2$ (conditional) = 0.37 | | | |

Males and females sniffed the pedal marks in 23% and 21% of their visits, respectively; while cubs, juveniles and undetermined bears did so in 61%, 48% and 26% of their visits,

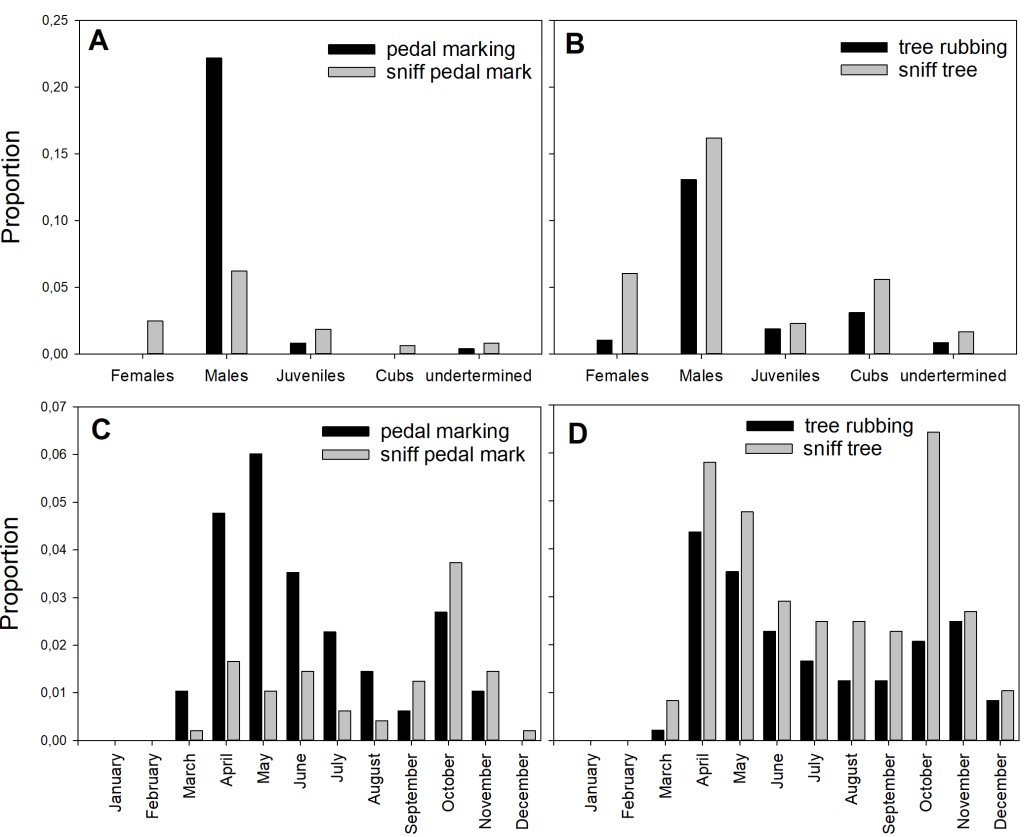

**Figure 2** **Proportion of the different behaviors.** Proportion of all observed behaviors performed by age and sex classes (A and B) and per month (C and D). Proportions were calculated as the number of observations within each class divided by the total number of observations of all behaviors in all size classes. Data in Tables S4 and S5.

respectively. The probability that a bear sniffed the pedal marks during a visit was higher outside the mating season (Table 3). Also, the lower the average precipitation and the average temperature in the preceding days, the higher the probability of sniffing the pedal marks (Table 3, Tables S3 and S8). Finally, the probability of sniffing the pedal marks tended to be negatively related to the time elapsed since the last time a bear performed pedal-marking at the site (Table 3; Tables S3 and S8).

Males performed pedal-marking in 81% of their visits to the site. They both pedal-marked and sniffed the pedal marks in 20% of their visits. Juveniles and undetermined bears performed pedal-marking in 17% and 7% of their visits, respectively, while females and cubs never pedal marked. The probability of performing pedal-marking by male bears visiting the site was positively associated with tree-rubbing by the same individual and negatively with the time elapsed since the previous visit of a bear that pedal-marked at the site (the shorter the time, the higher the probability of pedal-marking, Table 3; Tables S3 and S8). The association of pedal-marking probability with the remaining factors was weaker (Table S3).
Males sniffed the tree in 59% of their visits, while adult females did so in 51% of their visits. Cubs, juveniles, and undetermined individuals showed interest in the tree, sniffing it in 61%, 48%, and 26% of their visits, respectively. Interestingly, the probability of sniffing the tree by a visiting bear was higher the longer the time elapsed since the previous tree-marking event and negatively related to the precipitation during that period (Table 3; Tables S3 and S8), and was not affected by the sex or age class of the individual. Nevertheless, the model was not very explanatory (Table 3).

Males performed tree-rubbing in 48% of their visits. They engaged in both pedal-marking and tree-rubbing during the same visit on 43% of their visits and tree-rubbing and tree-sniffing in 35% of their visits. Adult females rubbed against the tree in just 9% of their visits. Juveniles, cubs and undetermined individuals tree-rubbed on 39%, 34% and 14% of occasions, respectively. Adult males and juveniles had higher probabilities of tree-rubbing during their visits than females (Table 3; Tables S3 and S8). The probability that a bear performed tree-rubbing during a visit was positively associated with tree-sniffing and pedal-marking by the same individual (Table 3; Tables S3 and S8), and tended to be positively associated with the time since the previous tree-rubbing event (Table 3; Tables S3 and S8).

Several recognizable individuals visited the site repeatedly (Supplemental Information), some of them throughout the study period. Four adult males visited the site between 10 and 35 times during the study, with up to 15 visits in one year (M1 to M4, Table S6). These males were frequent markers; for example, M2 and M3 were responsible for most of the instances of pedal-marking (59%, Table S7), while M2 was the bear that most frequently displayed tree-rubbing behavior (43%, Table S7). Additionally, other males visited the site sporadically (Table S9). These additional males were known individuals that were repeatedly observed near the study site (at least four additional males in 2012, five in 2013 and 2015, and seven in 2014). A minimum of one female visited the site in 2013 and 2015, two in 2014 and three in 2012. The minimum number of different individual bears visiting the site per year ranged between 11 in 2013 and 18 in 2015 (Table S9).

## DISCUSSION

In this work we show that the chemical communication behavior of brown bears at tree-rubbing sites is more complex than previously recognized, with pedal-marking being an integral part of this communication system. These marking sites form communication hubs where multiple individuals share and receive important information at the population level (*Sergiel et al., 2017*). Tree-rubbing is a well-known scent-marking behavior performed by bears (*Green & Mattson, 2003*; *Clapham et al., 2012*; *Sato et al., 2014*; *Seryodkin, 2014*; *Tattoni et al., 2015*; *Lamb et al., 2017*). Brown bears vigorously rub their flanks and back against the tree to scent mark it with secretions from the glands located on their back (*Tomiyasu et al., 2018*). They also mark other types of objects in the same way, especially in areas where the availability of trees is low (*Seryodkin, 2014*). Our results, in accordance with published information, show that tree rubbing can be performed by any class of individual at any time, but it is clearly monopolized by adult males, especially during the mating

season (see also *Clapham et al., 2012*; *Lamb et al., 2017*). Additionally, our results indicate that the information is received by all types of individuals irrespective of their age or sex.

Interestingly, tree-marking does not occur in isolation. Pedal-marking by males occurs as part of the marking process in association with tree-rubbing. As with tree-rubbing, pedal-marking is performed by males with a higher frequency during the mating season, while all classes of individuals act as receivers of the information. The existence of deep footprint marks forming one or more trails in the ground leading towards trees has been known for a long time, though not examined in detail (e.g., *LeFranc Jr et al., 1987*; *Clapham et al., 2013*; *Seryodkin, 2014*). Additionally, the typical behavioral sequence performed by males during pedal-marking has also been described with a variety of names, including bear dance, sumo walking, cowboy walk or stomping (*Sergiel et al., 2017*), but has been often interpreted as part of a stereotyped behavior leading to marking the tree and not a marking in itself. The recent description of pedal glands in the feet of bears and the concomitant pedal-marking (*Sergiel et al., 2017*) together with our results on the relationship between both pedal- and tree-marking provide new insights into scent-marking system in brown bears.

The data used in our description have some shortcomings that need to be considered. We provide data from only one site, although for a long period of nearly continuous monitoring. The area covered by the camera trap recorded only part of the area and, therefore, we may have missed behaviors, such as pedal-marking or sniffing when animals were out of the field of view; or tree-marking when the bears used other trees (there were nearby trees also used for marking). We could only detect sniffing behaviors when they were apparent in the videos, whereas bears have a very efficient olfactory system that might allow them to detect markings with little effort. Additionally, the zenith position of the camera trap may have limited our capacity to detect other potential marking behaviors such as urination or more complex stereotyped behaviors associated with tree-rubbing (*Clapham et al., 2014*). Despite these limitations, we believe that our results are relevant to the interpretation of chemical communication at marking sites by brown bears.

## Sending and receiving information

The importance of chemical communication at the site varied as a function of the individuals, depending on their sex, age, and presumably other conditions such as dominance or breeding status. Nearly half of the visits to the marking site were made by animals identified as adult males. They were responsible for most pedal-marking, and, to a lesser extent, tree-rubbing behaviors. Both behaviors were strongly associated when performed by adult males. Some males visited the site very often while others were more sporadic. Interestingly, some males marked in most of their visits while others mostly acted as information receivers. This may reflect a structure of dominance in the males sharing the area. Females, on the other hand, never pedal-marked and rarely rubbed the tree, and neither did the cubs accompanying their mothers. Young animals (of unknown sex) showed an intermediate pattern between males and females. Tree-rubbing was more frequently displayed by bears which also sniffed the tree and performed pedal-marking and positively related with the time elapsed since a previous tree-rubbing event, typically describing the

behavioral sequence of visiting males. Male brown bears have seasonally enlarged sebaceous glands on their back and prominent eccrine, apocrine and sebaceous glands in their feet; glands that are more active during the mating season, in association with their increased testosterone levels (*Sergiel et al., 2017*; *Tomiyasu et al., 2018*). Therefore, males acted as main sources of chemical messages at the site, as has been shown in other study areas (*Clapham et al., 2014*; *Lamb et al., 2017*).

Sniffing behavior, especially that of ground marks, is less obvious and therefore more likely to go unnoticed in videos. Nevertheless, all types of individuals showed interest in the chemical marks, acting as genuine information receivers. The probability of sniffing the marks during a visit was affected by weather conditions, with higher temperatures and precipitation in the preceding days reducing the probability of sniffing ground marks, a pattern that was not associated with actual pedal-marking, and higher precipitation negatively affecting tree-sniffing. The diluting effects of precipitation and temperature on the volatility of the odorous molecules left by bears at the marking site are a possible interpretation of these results. Interestingly, the probability of sniffing the tree was higher the longer the time elapsed since the previous visit, while it was the opposite for ground sniffing, suggesting a differential detectability between the chemical compounds secreted by pedal and back glands and among different substrates.

## Why brown bears visit these sites

Brown bears use chemical marking to convey information from senders to receivers. Why they do this and what type of information is transferred is still a matter of discussion. The chemical profiles of pedal and shoulder secretions indicate that they contain information on at least the sex and reproductive status of the individual (*Sergiel et al., 2017*; *Tomiyasu et al., 2018*). Additionally, it would not be surprising if information on the actual individual is also provided, as seems to occur with secretions from anal sacs (*Rosell et al., 2011*; *Jojola et al., 2012*). In species that normally exhibit a solitary non-territorial use of space, knowing the individuals whom they may encounter is quite valuable. Several non-exclusive hypotheses have been proposed to explain scent-marking in brown bears: self-advertisement for mate attraction, communication of individual dominance, competitor assessment and infanticide avoidance, with different roles depending on bear density (*Clapham et al., 2012*; *Lamb et al., 2017*). Our results show that chemical communication in brown bears is complex. Males are the main senders and also the main receivers, with some of them marking a lot while others tend to mostly receive information, indicating communication of individual dominance and the ability to assess male competitors. Male bears mark all year round but with a main peak during the mating season, a period of intense competition. This pattern has also been found at rubbing trees, both natural and artificially created to collect bear hairs (i.e., tree hair traps), in different ecosystems (*Green & Mattson, 2003*, *Karamanlidis et al., 2010*; *Sato et al., 2014*; *Berezowska-Cnota et al., 2017*; *Lamb et al., 2017*).

Females seem to visit the site less often, but all year round, and when they do, they are especially interested in receiving information. Knowing which males are moving around and their social dominance is very important for females in mate selection, since mating with the more dominant males that are present all year round would minimize the overall

risk of infanticide to their litters. Additionally, females with cubs of the year may benefit from knowing if a new male enters the area (*Bellemain et al., 2006*). Although more rarely, females, juveniles and cubs also rub trees, but it is unclear why they do it. In the case of juveniles learning by imitation may be the main reason (*Clapham et al., 2014*). Given that the sebaceous secretion in the shoulder of males is linked to testosterone levels, the secretion of females, cubs and juveniles can be expected to be testimonial or simply non-existent. If that is the case, their tree-rubbing may serve the purpose of masking their odor with that of adult males roaming the area. The resulting increase in chemical similarity could help to reduce the risk of infanticide by scent-matching (*Gosling & McKay, 1990*). If this interpretation is correct, tree-rubbing would have a scent-marking purpose only for males, while helping females and cubs to obtain a chemical camouflage by scent-rubbing as well as transitionally being part of the learning process of juveniles. In summary, there is no single best hypothesis to explain the role of these communication hubs, with the most plausible being a complex combination of dominance, mate selection, competitor assessment, mate selection and infanticide avoidance.

**Brown bear communication hubs**

Undoubtedly, sites like the one we monitored are important for brown bears at the population level. Our results show that the tree and the trails leading to it form a communication hub that most bears living in the area use to share and obtain information. Bears were the most frequent visitors to our site despite the easy accessibility and the fact that bears are not the most common large mammal. Bears choose specific trees in places that are well situated for the passage of other individuals (*Green & Mattson, 2003*; *Sato et al., 2014*). At these sites there is an association between different communication behaviors, with marking behaviors triggering the subsequent sniffing and marking of later visitors (*Berezowska-Cnota et al., 2017*). Nevertheless, these sites are not uncommon. In the vicinity of our site there were other trees used repeatedly by bears for marking (see Video S1). Brown bears maintain a dense system of marking sites that allow for a complex communication network over large spatial scales. Although they are not easy for humans to locate, several authors report varying densities of marking sites depending on bear density, including 0.26 sites/km$^2$ in the Italian Alps, 0.4 sites/km$^2$ in Hokkaido, Japan, 1.4 sites/km$^2$ in the Russian Komi Republic, 20 sites/km$^2$ in British Columbia, and 27 sites/km$^2$ in the Valley of Geysers on Kamchatka Peninsula (*Lloyd, 1979*; *Sato et al., 2014*; *Seryodkin, 2014*; *Tattoni et al., 2015*). Many of these studies describe trails evidencing pedal-marking (e.g., *Clapham et al., 2013*; *Seryodkin, 2014*). There are open questions that remain to be answered, such as the heterogeneity in the use of the multiple marking sites available to brown bears within their home ranges or the variability in marking intensity within and across populations.

## CONCLUSIONS

We showed that pedal-marking and tree-rubbing are strongly associated in a complex chemical communication system. At our site, bears visited more frequently during the mating season. More dominant male bears typically sniffed the depressions where animals

pedal marked, performed pedal-marking, sniffed the tree, and rubbed against the trunk. Adult males monopolized pedal- and tree-marking. Adult females, on the other hand, never pedal marked, and juveniles rarely did so. Females acted more as information receivers, rarely rubbing the tree. All sex and age classes performed pedal- and tree-sniffing, thus obtaining information on previous visitors. Different behaviors tended to occur during the same visit and were more likely if another individual had recently visited, generating long-term marking sites. These sites act as communication hubs that brown bears use to share and obtain important information on the animals present over a wide area at the population level. The intensive use of these sites and their number and density provide an idea of the importance of this communication system for this wide ranging, non-social large carnivore, with a complex mating system.

## ACKNOWLEDGEMENTS

We want to thank Aquila M. Pérez 'Kiti' for her help and support during the field work and Miguel Delibes for his help in the initial design of the study and his support during the analyses and writing.

### Funding

Eloy Revilla, Alberto Fernández-Gil, Nuria Selva, and Javier Naves were supported by projects CGL2017-83045-R AEI/FEDER EU, by the Agencia Estatal de Investigación from the Ministry of Economy, Industry and Competitiveness, Spain co-financed with FEDER to Eloy Revilla; Project 20166422 from the Principado de Asturias–Consejería de Agroganadería y Recursos Autóctonos to Eloy Revilla, Javier Naves and Alberto Fernández. Nuria Selva and Agnieszka Sergiel were supported by the National Centre for Research and Development (GLOBE, POL-NOR/198352/85/2013) and by the National Science Centre in Poland under projects BearConnect BiodivERsA 2016/22/Z/NZ8/00121 and project DEC-2013/08/M/ NZ9/00469. The publication fee was paid by the CSIC Open Access Publication Support Initiative through its Unit of Information Resources for Research (URICI). The funders had no role in study design, data collection and analysis, decision to publish, or preparation of the manuscript.

### Grant Disclosures

The following grant information was disclosed by the authors:
CGL2017-83045-R AEI/FEDER EU.
Agencia Estatal de Investigación from the Ministry of Economy, Industry and Competitiveness, Spain co-financed with FEDER.
Principado de Asturias–Consejería de Agroganadería y Recursos Autóctonos: 20166422.
National Centre for Research and Development: GLOBE, POL-NOR/198352/85/2013.
National Science Centre in Poland: BearConnect BiodivERsA 2016/22/Z/NZ8/00121, DEC-2013/08/M/ NZ9/00469.

CSIC Open Access Publication Support Initiative through its Unit of Information Resources for Research (URICI).

## Competing Interests
The authors declare there are no competing interests.

## Author Contributions
- Eloy Revilla and Alberto Fernández-Gil conceived and designed the experiments, analyzed the data, prepared figures and/or tables, authored or reviewed drafts of the paper, and approved the final draft.
- Damián Ramos Fernández conceived and designed the experiments, performed the experiments, prepared figures and/or tables, authored or reviewed drafts of the paper, collected the field data, and approved the final draft.
- Agnieszka Sergiel and Nuria Selva conceived and designed the experiments, authored or reviewed drafts of the paper, and approved the final draft.
- Javier Naves conceived and designed the experiments, analyzed the data, prepared figures and/or tables, authored or reviewed drafts of the paper, and approved the final draft.

## Animal Ethics
The following information was supplied relating to ethical approvals (i.e., approving body and any reference numbers):
   The fieldwork consisted in remote surveillance with a camera trap. This type of work does not require approval from a bioethical committee.

## Field Study Permissions
The following information was supplied relating to field study approvals (i.e., approving body and any reference numbers):
   Data was collected by one of the authors (Damián Ramos Fernández) during his work for Principado de Asturias–Consejería de Agroganadería y Recursos Autóctonos. Data access and use were granted by this agency (request CD0C2600000197400-33005G and exp-no. 2016/033072), and Damián Ramos Fernández was authorised to participate by exp-no. 2016/033072, Principado de Asturias-Consejería de Hacienda y sector Público.

## Data Availability
   Raw data are available in the Supplemental Files.

## Supplemental Information
Supplemental information for this article can be found online at http://dx.doi.org/10.7717/peerj.10447#supplemental-information.

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
