# Peer review of "Brown bear communication hubs: patterns and correlates of tree rubbing and pedal marking at a long-term marking site"

_PeerJ, doi:10.7717/peerj.10447_

## Round 0.1 · original submission · Major Revisions

Overview

This study reports observations of visitation and scent marking by brown bears at a single tree in the mountains of northwestern Spain over a nearly 4-year period. The study provided evidence, not always statistically supported, that visitation rate to the site varied seasonally, that the majority of visits involved behavior interpreted as providing and/or receiving chemical communication, that adult males were the most frequent visitors, that they performed both pedal marking and tree rubbing more than other sex and age classes and that they performed pedal marking more often than tree rubbing, whereas other size and age classes showed the opposite trend. In contrast to marking, sniffing marked areas was less frequent in adults than in juveniles and cubs, occurred in a similar proportion of visits by adult males and females, and was directed toward the tree more often than toward the pedal marks. The probability of a visitation was higher when the time since a previous visit by a male bear, the time since a bear had pedal marked, and the time since a bear had tree marked were lower. The probability of pedal marking and tree marking during a visit were positively associated. The probability of pedal marking was negatively associated with time since previous visit but positively associated with tree marking by the same individual. The probability of sniffing pedal marks was negatively related to time since the last visit and last pedal marking but higher outside the breeding season and after drier and cooler weather. However, the probability of sniffing tree marks was positively related with time since the last tree marking and higher after drier weather. Individual differences in visitation rate and behavior were also noted. Not all these patterns are addressed in the Discussion. The authors conclude that pedal and tree marking are parts of a complex chemical communication system, that adult males are the primary signalers, that adult females are primarily signal receivers, and the marking sites for communication hubs.

The reviewers indicated several major concerns with the validity of the findings, namely that marking was recorded at only a single site and that there was no validation of the assumption that individuals, age class and sex were correctly identified. I have some additional concerns that were not mentioned by the reviewers. Both reviewers and I find that there are numerous problems with grammar and word use.

As indicated in my comments below, I feel that detailed data from a single site can provide some valid conclusions. However, valid identification of age and sex classes is critical, and the paper cannot be accepted without reasonable support for this assumption. I have proceeded to provide detailed feedback on the manuscript in the hope that this issue can be satisfactorily resolved.

Editor's Comments

Title
Because you are only looking at a single site and are recording individuals whose home ranges and interactions are unknown along with their identities in some cases, it doesn't seem that you have the data to document what I understand by a 'communication hub'. I suggest removing the words up to the colon so that the title matches the contribution more closely.

Abstract
L24. Not clear what you mean by 'critical' and what sort of evidence there is for this statement. The statement differs from the cited arguments in the first paragraph of the
L31ff. After confirming the validity of the conclusions (see below for issues), review the statements of results in the Abstract to include only ones with evidence.
L43. I do not think that your manuscript provides the data to support this conclusion. It seems more of a hypothesis. Replace with a statement of conclusions and/or suggestions that are more directly related to the novel contribution of your study.

Introduction
L68-74. The concepts of synergy and communication hubs are frequently used in your manuscript. It is critical that readers know clearly what you mean. Although I am a behavioral ecologist, I cannot say that either concept is clear to me in the sense that I could define it unambiguously or recognize a pattern in communication data that did or did not support the hypothesis that a particular system was an example of either phenomenon. If you include these concepts in your manuscript (and I have some concerns about this as mentioned above and below), you must be much clearer on making them more than vague terms.
L91-94. It is critical for you to identify the knowledge gap that is addressed by your study. In particular, since your study provides an intensive examination of a single site, you could, if possible, emphasize topics that can be addressed more effectively by a continuous, long-term study one site than by shorter studies at multiple locations. These lines address only one aspect of your study, pedal marking. If it is really true that no previous study has examined age and sex classes of pedal marking or the environmental correlates, you should make a much clearer statement. However, you also need to be clear in the preceding literature review what is already known about the other main topics that your study addresses such as age, sex and season of visiting marking site and frequency of tree marking. This is where you would also identify what is known about synergy and hubs in marking, if you choose to retain these concepts. In the next paragraph you refer to a more complex communication system than previously acknowledged, but you haven't provided references for this lack of acknowledgement. More broadly, you need to consider whether only literature on brown/grizzly bears is relevant to your review or whether any similar patterns in other bear species need to be included (see my more detailed comments below regarding the Discussion).
L96. This paragraph defining your broad and specific objectives needs major rewriting. Can simply looking at whether bears use tree and pedal marking on the same visits provide evidence for 'a more complex communication system than previously acknowledged'? I don't see that the paper currently develops this hypothesis as implied. What evidence could you find to refute this hypothesis if what you are trying to do is not develop the hypothesis but examine the amount of support for it? What does 'identifying synergies' mean in empirical terms? Be sure that all the objectives are actually ones that can be and are addressed by your empirical data.

Methods
L126 The study is based on a single location. There is no attempt to justify the decision to put so much effort into one site and no indication of whether the authors consider it to be unique or typical. Both reviewers questioned the validity of this approach and I concur that it is not ideal. The authors must be very careful not to overgeneralize their findings. However, I think that it is possible to draw some valid conclusions about what happens at this particular site, so I am not willing to reject the manuscript on these grounds.
L134 I assume that this was a motion-sensitive video camera, but you don't state that explicitly. Did you obtain records in low light such as dawn, dusk and night?
L149ff. The analysis assumes that age classes of nearly all bears and the sexes of adults could be correctly identified. Some additional conclusions are based on recognition of individuals. While some general criteria are provided, the authors present no evidence of the accuracy of their categorization or individual identification. Furthermore, there is a risk of circularity if certain unacknowledged behavior (such as tree marking) influenced the attribution of sex to an individual. Both reviewers, experienced with this species, questioned this aspect of the methods. It is a critical issue in the validity of the study. Without it, the manuscript would be reduced to a study of visitation and marking rates by bears in general. One possible solution might be for a non-author researcher to use photographs or videos of individuals of known sex and age from studies by other researchers in other locations and quantify the success of the observer in identifying the appropriate class.
L166ff. The behavioral definitions for sniffing need improvement. It does not help a reader to define sniffing pedal marks as sniffing pedal marks (L171). I presume that on the video, you could not identify movements of the nostrils and are basing sniffing on larger body movements. For your study to be repeatable by other researchers, you should indicate what specific behavior patterns you used.
Also, in the Results you sometimes replace 'sniff' with 'smell'. Note that sniff is the behavior of inhaling whereas smell is the perception of odor. Therefore, you could potentially detect when an animal sniffs but not when it smells.
You should also recognize that bears can presumably smell conspecific odors without the specific behavior of sniffing, so there may be substantial perception without a clear visual signal.
L175. Although you state that you recorded the number of behavioral events per visit, the data presentation is based only on whether or not each type of behavior occurred in a visit.
L179ff. The statistical analysis section of Methods may require changes as indicated by my comments on the Results below.
L181. In reading the Results, I found unexpected variation in the independent variables for each category of dependent results. I think that a brief justification for the choice of variables to examine would be appropriate.
L187ff. It appears to me that you are mixing the information theoretical approach with classical statistical probabilities. Although I am not an expert in statistics, my understanding is that this is incorrect. Please check with a statistician to be sure that your statistical approach and decisions regarding what results to present is appropriate.

Results
L199. The number of other species detected is unrelated to any of the questions addressed by the manuscript. I think you should remove Fig. 1 and simply list the common and scientific names of the other species, indicating that visits of all of them were less frequent than those of bears, although some of the species were more common in the area. (It might be appropriate to mention which ones were more common.)
L200ff. The Results are quite challenging to follow and need substantial reorganization. I think it would be clearer to organize by the behavioral categories (visits, pedal marking, tree marking, pedal sniffing, tree sniffing), incorporating the univariate comparisons along with the relevant model results into each to provide evidence of effect sizes and statistical support together. To improve clarity, it is important to present the comparisons in consistent order. This applies to male vs. female, pedal vs. tree marking, marking vs. sniffing. For example, on L225-228, you refer to male frequency of pedal marking and sniffing but for females the frequency of sniffing and pedal marking. Sometimes you present marking before sniffing (L222, Table 2), but in other places (e.g., Table 1, model results), sniffing precedes marking. This makes it very difficult for readers to grasp the patterns.
Visitation.
The sex and seasonality of visits seem important but are incompletely developed.
• The overall proportion of visits by each age and sex class (L200ff) is a good start, but the reader will want to know if the sex difference is statistically significant, so you should look for a way to test this. A simpler univariate comparison would probably be a useful complement to the more complex model. (This also applies to the other behavioral categories.)
• I suggest that for each age and sex class you provide the median and range of visits per day, including only days on which the camera was functioning for the whole day. Figure 2 is not very clear because it appears that visit rate per day is confounded by the camera function in the first couple of years. You should still include the data on whether the camera was functioning.
• Information on seasonality is scattered through the results and not statistically well supported.
o Although you note that most visits were in spring and summer, you do not provide evidence that visitation rates were significantly different during this period.
o Your GLMM for visitation apparently compared breeding to non-breeding period, but the non-breeding period included some months of high visitation rate. Is it possible to divide the entire year into biologically relevant seasons and ask whether the differ?
o I don't think that you have indicated what a positive or negative sign on seasonality means; be sure that this (and any other categorical variables such as weather) are clear to readers.
o Is there a hibernation season that should be excluded from seasonal comparisons?
o Are there enough data to examine whether sexes differ in their seasonality?
o There also appear to be some differences between years. Is there any indication that these could be real and potentially related to the environment. For example, some bear species are known to delay hibernation in mast years of their primary food source.
• For the effect of time since a previous visit, it seems that any effect would be confounded with seasonal variation in rate. During periods of higher visitation rate, most visits would necessarily follow a shorter interval. If this is the case, any inference that a previous visit might increase the probability a subsequent visit is greatly weakened.
• You don't mention anything about diel patterns of visitation. Do you have the data to say about whether visits were more frequent during the day or night? If your camera did not function at night, could you have missed some visits?
L217. I don't think that the proportion of all events in a particular category attributed to each sex is very relevant because it confounds the visitation rate and the behavior performance rate.
Behavioral events
• You should decide whether you will examine events per visit as stated in the Methods or continue to use presence or absence of an event on a visit. I don't have any problem with the presence/absence analysis, but you should be clear about what you are doing. Perhaps an indication of the range of number of events would be helpful to readers.
• You should examine whether the differences in probability of an event per visit among sex and age classes is statistically significant, at least for adult males and females.
• You should be clear on which independent variables apparently predicted an event and which did not.
• If you want to draw conclusions based on the occurrence of multiple categories of behavioral events occurring in the same visit, you should provide evidence that this occurs more frequently than expected by chance.
• If you want to draw conclusions based on individual differences in events per visit, you should provide evidence that these differences are statistically significant.
• Does the skewed distribution of visitation rate and differences in behavior among individuals require more consideration? Your analyses seem to assume that events are independent. Is there any pattern in visitation rate and behavior performed that might provide some insights into the functional role of the different behavioral events?
L253. In contrast to the text, Table 3 shows a positive effect of pedal marking.

Discussion
I have not provided detailed feedback on the Discussion because it needs major changes. Presently, it draws broad conclusions without provided detailed logical examination of the evidence supporting them. For each of the patterns you found, such as sex differences in marking behavior and seasonal patterns, you should provide a careful and critical evaluation of the evidence your study provided, conclusions that can be drawn, and how these relate to the existing literature. For broader topics such as the complexity of olfactory communication systems and trees as communication hubs, you should make sure that the concept is clear and indicate precisely what your study adds to the literature. The Discussion should be focused on topics to which you make a contribution rather than more general literature. For example, you did not measure density of marking sites but provide detailed values from the literature. On the other hand, differences related to sex, age, season and time since last visit receive very brief attention.

In providing a literature context, you should make a clear decision about whether to include literature on other bear species. If they behave very differently, you may not need to cite this work other than indicating that brown/grizzly bears are very different. However, if they are similar, it seems appropriate to cite literature on black bears or other species when you are providing evidence for brown bears that they are performing behavior more strongly established for other species. In this context, you might find interesting a small observational note by Thomas Reimchen on use of trails with deep footprints by black bears on Haida Gwaii in British Columbia Canada (Canadian Field Naturalist 112:698-699 (1998). This doesn't seem to be available online but is available at Reimchen's website at the University of Victoria, Canada.

L406. Have you justified how visitation rates on the order of 0.6/day with a peak of about 1 per day can be described as 'intense'?

Grammar
When you have made the appropriate changes in the manuscript, please find a native English speaker who can help revise the manuscript in detail. There are large numbers of small grammatical and word-use errors that need to be corrected. These are far too numerous for me to note, but I have provided a detailed annotation of the Abstract on the attached manuscript to give you an idea of the magnitude of the need. In other places, I sometimes highlighted words needing attention but did not attempt to provide corrections or a complete list of problems.

Additional comments

L92,104 and elsewhere. I suggest using the term 'sex and age classes' rather than the hyphenated form and being consistent throughout the paper in whether sex or age comes first when referring to both.
L131 'Data' is plural.
L139. I think that stating that the camera was placed vertically above the marking location would be clearer than 'zenithal', a word which is rarely used in English. I think it would be useful to indicate the dimensions of the area of ground covered by the field of view.
L197. Visits should be defined in Methods, not Results.
The text associated with the video also needs improvement. Was the video really taken through a spotting scope rather than a camera with a telephone lens?

Reviewer 1 ·

Basic reporting

1-The article could be improved by editing for proper English.
2-The study does a good job of including relevant literature for a thorough background of the topic.
3-The article structure is good.
4-The article is self-contained.

Regarding #1, overall, the article seems like it could be improved by a thorough editing by a native English speaker. For example in the abstract:
Line 29-30: I don't understand what "automatic video recording" means. Is this a video camera trap?
Line 31: "visited more frequently" should be "more frequently visited".
Line 32: "consisted in sniffing" should be "consisted of sniffing"
Line 41-42: I don't know what "These synergies consisting" means.
You can likely get this paper through the review process as is, because these are just minor grammatical errors, but your manuscript will be taken more seriously by other scientists if you address these type of issues throughout the manuscript.

Experimental design

The experimental design is interesting and solid in design, and the methods are explained well. The main drawback is that the study is limited to only one site, but this drawback is addressed thoroughly in the Discussion.

Validity of the findings

1-PeerJ does not assess the novelty of studies.
2- The study is limited to one marking site.
There is also a lot of interpretation used to sex and age the bears (i.e., head and neck shape), and this lack of rigour should be mentioned in the Discussion.
In addition, at times there is no distinction made between bear species, which I think could be a week part of the manuscript. For example, brown and black bears may have different behaviors based on their ecological niches.
3- Discussion and conclusions are well-written and accurate.
4- Speculation is limited.

Additional comments

This is an interesting and well-done experiment. As noted above, I encourage you to have a native English speaker proofread the manuscript to catch small grammatical errors.

Reviewer 2 ·

Basic reporting

1) The paper needs a hypothesis
5) The English needs to be edited in this manuscript

Experimental design

2) The analysis essentially only has one degree of freedom. A single tree is used. Because of this, the paper may be better suited for a natural history, or research note type of publication.

Validity of the findings

3) Visually identifying individual identity, sex, and age, needs to be validated, or the authors need to provide references where others have done this successfully for bears and have validated it. Without this, I am not confident that visually identifying individual identity, sex, and age of bears is a robust approach. Pictures showing how and why animals were identified as such would help, as would exploring the sensitivity of results to this uncertainty.
4) Many studies of brown bears in North America detect similar numbers of females (identified via DNA genotyping) and males at rub trees (Kendall et al. 2009; Morehouse & Boyce 2016; Lamb et al. 2018). It should be acknowledged, that the results found here (9% of females rub) are either not consistent with North America, or are an artefact of only sampling a single tree.

Additional comments

This manuscript provides an interesting account of the function of pedal marking and chemical communication by brown bears at a single oak tree in Spain. The following items require attention before this work is ready for publication
1) The paper needs a hypothesis
2) The analysis essentially only has one degree of freedom. A single tree is used. Because of this, the paper may be better suited for a natural history, or research note type of publication.
3) Visually identifying individual identity, sex, and age, needs to be validated, or the authors need to provide references where others have done this successfully for bears and have validated it. Without this, I am not confident that visually identifying individual identity, sex, and age of bears is a robust approach. Pictures showing how and why animals were identified as such would help, as would exploring the sensitivity of results to this uncertainty.
4) Many studies of brown bears in North America detect similar numbers of females (identified via DNA genotyping) and males at rub trees (Kendall et al. 2009; Morehouse & Boyce 2016; Lamb et al. 2018). It should be acknowledged, that the results found here (9% of females rub) are either not consistent with North America, or are an artefact of only sampling a single tree.
5) The English needs to be edited in this manuscript
Abstract: “consisted in sniffing” to “consisted of sniffing”
“Capitalized” to “dominated”?
“more likely if other individual” to “more likely if another individual”
“obtain important information at population level”- what kind of information? How do you know?
L51-mating or finding and attracting mates?
L57-60- Urine doesn’t provide information on reproductive status?
L73- “in” to “to” and “at population level” to “at a population level”
L86- the other paper I know of for this is “Lamb et al. 2017, Plos One” which is cited elsewhere in the paper, but could be added here.
L96- make the hypothesis clear here. Do you expect Pedal marking to be additive to tree rubbing? Ie. Same information transferred, but just more effectively, or, does the pedal information interact with tree rubbing information and together they paint a picture?
L102- how did they communicate before 2002? Or were they few in number before then?
L114- what is the population density? In bears/1000km2
L122- choosing only a single tree is surprising, this severely limits effective sample size and inference.
L139- define zenithal in this context. Is this looking down from the tree onto the rubbing area? And when was this change made?
L149- I would like the authors to add additional information on how they visually identified individuals, sex and age-class. Specifically, an analysis of the classification accuracy and repeatability of visual identification would be insightful, especially within and between years. Other works have visually identified bears, notably Larry Aumiller, who ran McNeil Falls for 25 years could supposedly do this. But he watched the bears every day from a close range. A lot was by how they walked, how they fished, who they got along with. He knew the bears all very well. I do not think that individual identity, sex, and age of bears can be reliably determined from camera trap footage of bears (even though others have done this, but with no validation as to whether the classification was correct [Clapham et al. 2012]). I would like to see numbers regarding the accuracy of these methods. Especially because large classification error rates could significantly change the results of the current manuscript.
L265- is it possible that animals are still receiving signals even when you don’t see them sniffing? Maybe they only need to sniff when the signal is weak- after time has elapsed.
L328- discuss the sex and age classes most likely to receive these signals. What are they supposedly interpreting from these signals? Do these patterns change through the seasons?


References:
Kendall, K.C., Stetz, J.B., Boulanger, J., Macleod, A.C., Paetkau, D. & White, G.C. (2009). Demography and Genetic Structure of a Recovering Grizzly Bear Population. J. Wildl. Manage., 73, 3–17.
Lamb, C.T., Mowat, G., Reid, A., Smit, L., Proctor, M.F., McLellan, B.N., et al. (2018). Effects of habitat quality and access management on the density of a recovering grizzly bear population. J. Appl. Ecol.
Morehouse, A.T. & Boyce, M.S. (2016). Grizzly bears without borders: Spatially explicit capture recapture in southwestern Alberta. J. Wildl. Manage., 80, 1152–1166.

---

## Round 0.2 · Major Revisions

Revision has substantially improved this manuscript. However, the reviewers and I still find some problematic issues. Some of my concerns and those of a reviewer relate to the statistical analysis. Because I lack the expertise to judge these issues, I sent the review out to a third reviewer, a biostatistician, with the request to focus on statistical issues and a list of my particular concerns.

Reviewer 1 suggests that is still needs major revision. Reviewer 2 still suggests rejection, but provides detailed comments that could improve the manuscript. These criticisms do not indicate that the manuscript is not publishable and may reflect a perception that the paper should be rejected then invited for resubmission. Thus, I am treating this review as if it requested major revisions. The statistical review indicated that a number of major issues remain. Therefore, my decision is that major revision is still required.

In the following comments, I first provide additional comments on some of the issues raised by the reviewers, then add some suggestions of my own. Because the statistical issues are critically important, I address those first. At the end of my comments, I have provided the text of my request to Reviewer 3.

Editor’s Comments on the Reviews
Reviewer 3
This reviewer has provided a very clear summary of the problems along with detailed guidance on possible solutions. Please take this advice seriously. I recognize that some of the solutions may result in loss of statistical significance. I am prepared to accept a manuscript describing the findings, even if some of the patterns are not statistically significant, as long as the analysis is correctly done and the conclusions match the statistical results.

Reviewer 1
English use. Grammar and word use are much better than in the previous version. I have provided an annotated pdf that has additional corrections of word use and punctuation.

Note that either British or U.S. spelling are acceptable to PeerJ, but they should not be mixed. I found inconsistencies in the spelling of behavior/behaviour, meter/metre, characterize/characterise and others. Please set your entire document in one spelling convention and check the entire text, including tables and figure captions (except, of course, the references) for consistency.

Use of term ‘bears’. In the paragraph of the Introduction starting on L77, I interpret ‘bears’ as referring to ursids in general following from the topic sentence. If you are already referring only to brown bears, you need to clarify. The paragraph starting on L86 seems clearly to narrow the focus to brown bears. However, the reviewer is correct that the reference to unspecified bears on L91,92 creates ambiguity. The easiest clarification would be to replace ‘bears’ with ‘brown bears’ throughout the manuscript. Alternatively, you could modify L87 “. . .Ursus arctos (hereafter bears) across its Holarctic range, . . .” Whenever you subsequently refer to another species, you would have to be explicit.

I agree that using ‘camera trap’ consistently will reduce ambiguity.

I agree that the ‘synergy’ needs a bit of explanation. I believe that the reviewer has misunderstood your meaning. My understanding is that synergy refers to more reliable communication using two or more signals or modalities rather than cooperative communication among individuals. However, this difference in our interpretations shows the need to indicate more explicitly what you mean.

I agree that L103 would be an appropriate place to offer a bit more explanation about the origins of a multiyear data set at a single site. The manuscript makes more sense as an analysis of the opportunistic availability of a multiyear data set obtained with other goals than it does as a study designed a priori to investigate communication hubs. This is an example of the reviewer’s comment that you have clarified information for the reviewers and editor but sometimes did not clarify the issue for readers of the final article.

Reviewer 2
The reviewer suggests that you are ‘overselling’ your study by insufficient acknowledgement of the limitations. Although I did not have as much of a problem with this aspect as the reviewer did, I think that you should acknowledge more explicitly the limitations of the study at the same time indicating why it is advantageous to publish the analysis of the findings despite these limitations.

Acknowledging the limitations. The paragraph on L326-337 helps to make explicit the limitations of the study that were highlighted by the reviewer’s comments on ‘overselling’ and limited applicability (comment re L337).
a) With regard to using only a single tree, is there any evidence from other studies that marking patterns differ substantially or are very similar at different trees within the same general location? If not for brown bears, is there any such evidence for other species about local spatial variation in marking patterns? Might such variation be expected, based on what we know about olfactory communication? I think that it is important to briefly address the implications of studying only a single tree.
b) Also, as noted by the reviewer, bears in different regions might behave differently, and this should be noted.
c) Finally, you have provided evidence that relatively few bears contribute strongly to the data yet your analysis used visits as the units of analysis, as if they were statistically independent. This means that idiosyncrasies of particular individuals could strongly influence patterns in the data. This is a very important to acknowledge, and Reviewer 3 has suggested an approach to the problem. Although less than ideal because of incomplete identification of individuals, such an analysis will strengthen the manuscript.
d) With the limitations clearly acknowledged, I think that the final statement (L335-337) can remain as is; you are clearly referring to a personal belief or hope, not drawing a scientific conclusion.

With regard to the reviewer’s objection to the expression ‘long-term study’, I think that in the Introduction, where you introduce the data source or earlier in that section, you could indicate the duration of previous studies of bear marking, thus placing the duration of your study in context. (The reviewer variously refers to it as 4 or 5 years, but if I understand you correctly the data used spanned 3 years and 8 months of recording.) If your study is not ‘long-term’ in comparison to the longer studies, you could simply refer to it as a ‘multi-year study’. If inter-annual differences are notable, you could also put a bit more emphasis on the importance of collecting data over several years.

With regard to the ‘long-term marking site’, you could perhaps elaborate a bit on the description. You have evidence for use only starting 2002. At this time, was the site already well established, for example by deep foot prints leading to the tree? How recently has the site been confirmed to be still in use? I will leave it to you to decide whether the using long-term in this context could be misleading or informative for a reader but provide brief justification for your choice in the manuscript.

Identification: repeatability and accuracy. I was reassured by the detail provided regarding identification of sex and age classes and of individuals. An objective measure of the reliability of these researchers would be desirable, but I cannot think of an objective method to do this. I cannot see that asking less experienced individuals to try to identify individuals as suggested by a reviewer would provide such a measure. However, it would be possible to at least provide a measure of inter-observer reliability for the different technicians who classified the sex and age class and the individuals. One researcher could randomly select a set of videos that included cubs, juveniles, females and males (individually known and not known) and present them to each of the three observers with the identification role to record sex and age class and individual. Then a measure could be provided of the agreement among the researchers. Please indicate the number of researchers who classified the age-sex classes of the bears and identified individuals and include their initials in the manuscript. This is another case where you provided detail in your response document that would be useful in the manuscript itself.

L28. I suggest: “We describe the patterns of chemical communication used by different age and sex classes, including differences in the roles of these classes as information providers or receivers over five years at a long-term marking site.” Similarly, L102 “depending on the primary roles of these classes as either information providers or receivers.”

Hypothesis: The reviewer has a valid point about the testability of the hypothesis on L99. My perspective is that the hypothesis would be acceptable, if this is truly what you were interested in at the start of your study, providing it was combined with predictions that could clearly support or refute it. This could be done by modifying the second sentence in the paragraph to include what patterns might indicate that the hypothesis is incorrect. If there is no pattern that could refute it, the hypothesis is probably too vague or general and needs refinement. Such refinement could involve the more focussed hypotheses and predictions as proposed by the reviewer. However, you must be careful not to propose hypotheses after the study is done because this leads to logical circularity. If yours was basically a descriptive study with an interest in the variation among sex and age classes or other variables, I would rather see it presented as such rather than presented in the context of a ‘hypothesis’ developed after the completion of the study.

Statistics.
In my previous comments, I suggested providing a brief justification for your choice of variables. In your reply, you elaborated slightly on the brief mention of L211 but did not add the detailed justification. This is essentially what Reviewer 1 is also requesting. For example, it may be reasonable to include weather as a predictor only for sniffing behavior, but you should not leave it to the reader to have to figure this out. The choice of which temporal variables to use is less obvious. Also, it is not clear why sex and age class would not be included in the analysis of visits. Please explain your choice of variables. If the number of predictor variables is a constraint, you could include different predictors in different analyses to get at different questions.

I do not agree with the reviewer that you need to provide statistical support for the frequency of bear visits vs. other species. This is an extremely minor point and not related to any of your conclusions.

Regarding L225-227, I think what the reviewer is suggesting is that you provide a figure of the diel pattern of visits in the supplementary material. This seems like a useful suggestion.

Regarding L227-228, I agree that it would be useful to provide the data on means (or median if not normally distributed) values for the breeding and non-breeding seasons, rather than just the GLMM evidence that season is statistically significant. The data should be scaled so that the comparisons are valid, e.g., visits per operating camera day. Also, be clear that the non-breeding season values did not include the normal hibernation period.

Editor’s Additional Comments
L129. Five years does not agree with 3 yr and 8 mo in Methods.
L202. On the pdf, I suggested a rephrasing to conform to the fact that this is a correlational not experimental study.
L232. I don’t see why Fig. 1 is cited here.
Fig. 1. Why does the caption refer to 2013-2015 (three years) when 4 years are shown on the graph?
Fig. 2 would be much more useful as proportion of visits with each type of behavior and number of visits per recording day. Proportion of visits is what you can examine statistically and it has more generality for future researchers than the absolute number of records. Number of visits per recording day allows for the variation in camera function. Furthermore, I don’t think that what the present figure is technically a frequency distribution. Label the panels A,B,C and D, and write a full caption in complete sentences that parallels Fig. 1 and refers to each panel. I don’t see a need to refer to the supplementary material here. But if there is a good reason, please indicate why a reader should look at it. Capitalize the months as in Fig. 1.
Tables 1, 3. Please come up with more intuitive short expressions to describe the independent variables. It is very hard to understand Table 3 because the reader needs to keep referring back to Table 1. I suggest putting the description first and then the short term in parentheses. For example, ‘age-sex’, ‘age-sex previous tree marker’, ‘time since previous visit’, ‘time since previous male visit’. The behavioral variables seem as written. Also, you can condense the description using the form ‘log10 days’ with 10 as a subscript (assuming that the logarithm used is base 10).
Table 1. See notes on Table. Specify the base of the logarithm.

Editor’s request to Reviewer 3

Thanks again for agreeing to this task. I hope it doesn’t prove too onerous. I am providing this summary so that you don’t have to get too deeply into the manuscript and all the files in order to give me the help I need. As I mentioned, I am not looking for a full review, but an evaluation focussed on the adequacy of the statistical analysis.

This is the first revision of the manuscript. The authors have analyzed camera trap videos of European brown bears scent marking (producing communications) and sniffing scent marks (receiving communications) at a single tree. The camera operated almost continuously over several years. The authors are interested in differences among sex and age classes and between breeding and non-breeding seasons in the marking and sniffing. There are two types of marking, rubbing the back on the tree (tree marking) and rotating the feet on deeply indented tracks leading up to the tree (pedal marking). Sniffing can be directed toward the tree or toward the pedal marks. These are considered four classes of communication events. [The supporting video in the supplemental material is worth a look; it is quite neat and helps make these behaviour patterns clearer.] The Abstract presents the results as largely descriptive statistics concerning visits and behavior at the site, but the Results do draw some conclusions about relative rates of sex and age classes and temporal pattern of behavioral events.

In broad terms, I would like to be sure that the analysis used is appropriate, that it was correctly carried out, that the needed information is correctly presented in the tables, and that the authors have interpreted it correctly. The reviewers did not question the type of analysis but they have raised concerns about the presentation. I also have some questions about the analysis. I have tried to put specific questions in bold, but you should feel free to raise any issues you feel are important.

The authors used separate GLMMs for 5 dependent variables, the probability of a visit on a day the camera was working and the probability of each of the four behaviour events (sniff pedal, mark pedal, sniff tree, mark tree) occurring on a given visit. See L200-213. The variables chosen as potential independent variables varied among the five analyses according to what the authors considered biologically reasonable. For example, weather variables since the most recent visit are included only for sniffing. These are shown in Table 1. Table 2 provides the sample sizes for the different behaviours and different sex and age classes. The abbreviations for the various independent variables are not very intuitive. To save readers switching back and forth between Table 3, Supp A7 and Table 1, I will ask them for clearer acronyms.

One issue is the problem of (pseudo)replication. The authors were able to recognize some individual bears, especially males, and some of these visited multiple times. Yet, the authors treat individual visits as the statistical units. Because the authors were not able to recognize all individuals, the analysis cannot be redone using individuals as the units. I intend to allow this issue to remain but will insist that the authors provide appropriate caveats for their interpretation.

The authors do not mention anything about testing assumptions of their test.
• Should they have tested for normality and correlations among the independent variables?
• Do they have too many variables for their sample size?
• Are there any other assumptions they should address?

The first part of the Results presents the proportion of behavioral events of each type that were performed by the different classes (L243-262). This combines probability of a visit and probability of performing the behavior. I don’t consider it very important or in need of statistical analysis, although it is a prominent part of the results and abstract and a referee did request statistical analysis. I think I will allow it to remain descriptive. They can’t test frequencies against expectations because they don’t know the sex ratio or proportion of juveniles and cubs.

The authors do not test whether sex and age class influenced the probability of visits. It may be obvious why sex and age class could not be included in the independent variables for visits, but I cannot see it.
• Does it seem possible to include sex and age class as an independent variable for probability of a visit on a day as it is for probability of performing a behavioral act on a visit?

The key results for each of the four categories of behavior start on L255, with a separate paragraph for each.

If we taking sniffing pedal marks (L255-262) as an example, we can see some of the issues. In Table 3, they present what they call ‘estimates of the effects of the factors’, with SE and a p-value. I don’t see any analysis method for how the p-values were arrived at. I questioned whether they were mixing AIC and hypothesis-testing on the first version and they said:
“The approach we used is standard in model selection using AIC. We used a package specifically built in R (MuMIn) to select models within some level of support based on AIC. This approach has been used in multiple occasions in scientific publications, including PeerJ.”
• Is the inclusion of p-values correct, and, if so, do they need to explain it in more detail?
The Results for pedal sniffing lists the significant factors and one that was marginally non-significant by traditional alpha levels and doesn’t mention those that were non-significant. It isn’t obvious how the reader would know that a positive sign for season indicates more sniffing outside of the mating season.
• Should they somewhere indicate what a positive sign means for season?
Supp. Table A7 lists all models that were within the delta < 2. I presume they are using the two models where visits were included as the justification for the parenthetical conclusion that there was a negative effect of time since last visit, even though the p-value was high. A reviewer said that Table 3 needs standardized coefficients and needs to be model averaged. A reviewer questions how they can say that their paper is about age and sex effects when only one of the top models includes sex and age class for each of their analyses. Checking Table A7 indicates that this assertion by the reviewer is not correct; models with age-sex range from 0 to 3. Sex and age class are included in only tree rubbing in Table 3 and here they distinguish different classes. However, I am not sure what analysis they used to achieve this distinction. I wonder if having so many classes reduces the chance of finding an effect, whether a separate analysis to compare males and females would be worthwhile, and whether a post-hoc test is required.
• Are they correct in their analysis, interpretation, and presentation of required data?

Thanks so much for this.

Reviewer 1 ·

Basic reporting

Generally good English throughout, but could be improved.

Terminology could also be improved (referring to 'bears' and assuming the reader knows that it is brown bears rather than bears of the family Ursus)

Experimental design

Acceptable.
Descriptive in nature.
Could be improved with simple descriptive statistics, rather than just providing proportions/means.

Validity of the findings

Acceptable

Additional comments

Brown bear communication hubs: patterns and correlates of tree rubbing and pedal marking at a long-term marking site (#32708)

My main comment is that the analyses should become more rigorous, rather than purely descriptive.
-Analyses (Lines 200-213) and Table 1. Why are the age and sex of individuals not considered for each behavior? Even if some behaviors are performed by just males or adults, this would be important to analyze. Please use the age and sex variables for each behavior.
-Lines 221-222: Rather than saying that others species are “considerably” less frequent. Please analyze statistically and tell us if bears are significantly more frequent. (Make this explicit statistically rather than purely descriptive and lacking rigor).
-Lines 225-227: It would be helpful to plot the time of use in the overlap package and provide a figure.
-Lines 227-228: Again, rather than just saying they are more frequent visitors during breeding season, please analyze this statistically.
-Lines 243-284: rather than just reporting percentages of behaviors by each age and sex class, it would be helpful to actually analyze these to see if they are significantly different. This could be easily done with chi-square tests.

Other comments include:
-I would suggest using the term “camera trap” throughout the manuscript, rather than “camera”, “automatic camera” or other terms.
-You have made many clarifications to your responses to us (editors/reviewers), but these clarifications often have not made them into the manuscript.
-Your use of ‘bears’ to denote brown bears throughout the text is confusing. I think you should specifically mention the bear species at each instance of ‘bear’ rather than having the reader make assumptions about if you are talking about the family Ursus or a given species.
-Line 71. Please define “synergy”. I tend to view communication as a competitive behavior rather than a cooperative behavior. For example, two territorial male bears will leave scent to show their dominance and enhance their odds of being chosen to mate with by a female. I don’t think this is a cooperative behavior between the two males.
-Lines 102-106: Please justify why you used just one marking site. Is this site unique in some way? Or is it typical? You need to justify this for future readers of the article.

Reviewer 2 ·

Basic reporting

The paper still feels "over-sold" to me. There are many hyperbolic statements suggesting this is a "long-term" study, or that the results could inform what brown bears do generally. I do not feel that is the case at all. Monitoring a single tree for 4 years is neither long term nor broadly applicable.

Experimental design

The statistics need additional attention and analyses

Validity of the findings

I still feel this work needs much more qualifying of the results, due to the small sample, and unique situation of a single tree used by a recovering, low-density bear population.

I would also like to see the authors have a few independent persons try to identify individual bears based on their criteria. And calculate the repeatability and accuracy of this approach.

Additional comments

Title and L29: Long-term generally refers to studies of > 20 years. 5 years of data collection for a mammal that can live for >30 years would not be considered long term. All references to long term should be removed. Even if the long-term is referring to the long-term duration of this one tree by bears, that time is unknown (aside from atleast 2002-2016) so, also warrants removal or clarification as well.

L28, who is “their” referring to? The chemicals? The bears?

L99: I should have phrased comment about the hypothesis from the last round differently. Yes, the authors alluded to having a hypothesis, but it was vague, and arguably not falsifiable/testable. “hypothesize that pedal-marking and tree-rubbing are deeply linked, forming a more complex communication system than previously recognized” is a subjective, unfalsifiable claim as far as I am concerned. I would like to see a specific, mechanistic hypothesis. “we hypothesize that tree-rubbing are deeply linked because….we expect to find adult male biased marking behavior during X season because…” Then list your testable predictions.

L111: remove any plural, such as “these communication hubs” given that the authors only have a single tree. N=1.
L210: shouldn’t individual be a random effect to, due to the repeated observations

L228: Table 3 needs to be converted to standardized coefficients, and these need to be model-averaged, given that there are many competing models for each analysis.

L228 (table A7 Results of the models within ΔAIC < 2 of the best model): how do the authors justify their paper being largely about age and sex effects, when age and sex effects are only in 1 of the top AIC models for each of the 5 analyses.
L241: the 22% and 87% # below don’t make sense to me. In 22% of visits there was no communication behavior, but in 87% of visits there was chemical communication?
L263:296: I think there needs to be some statistical tests associated with all of these %’s reported. Are these comparisons statistically different? Ie., I suspect “Males sniffed the tree in 59% of their visits, while adult females did so in 51% of their visits” would be statistically indistinguishable.
L327: describing “January 2012 and January 2016” as a “very long time period” is a major stretch. I suggest just saying 4 years. Especially given that not all data during that period were even used

L337: suggest adding “ for brown bears living at low densities in Spain”. I don’t expect all of these results to be generally applicable across the brown bear range. Partly because I suspect the n=1 nature of this study will be sensitive and extrapolate poorly, but also that brown bears have different behaviours, life histories and stressors across their range that will influence even a solid inference.

·

Basic reporting

This review was requested as an additional, partial review to address statistical concerns, in particular several raised by the Academic editor.


Minor comment

It would be nice to see some of the main conclusions in the abstract.

Experimental design

Which package and version in R was used to calculate the GLMMs? Was it lme4? This should be stated and cited.

Validity of the findings

1. Problems with pseudoreplication
The data is definitely pseudoreplicated. As individuals are known to have made multiple visits, the sample size is inflated. I don’t think this is necessarily an insurmountable problem, but it needs to be addressed more than it has.

Either a really careful interpretation of the results needs to be made, or a solution (partial solution) needs to be attempted. One potential solution (not perfect but better than nothing) would be to add ID as a random factor [a crossed, not nested random factor, i.e. with lmer this would be (1|year) + (1|id)]. Identify the bears which can be identified, and identify all other visits as “unknown”. This would result in lumping all unknown visits together, erring on the side of being overly conservative (opposite to the current situation). Both options (not conservative enough, vs. overly-conservative) have their draw backs.


2. Age-sex class for visits
Why isn’t age-sex class included as a parameter in the model looking at the probability of a bear visit? One of the main, and potentially most straightforward conclusions is that adult males make more visits than any other class. It would be great to have this backed up by statistics.


3. Analysis and Information Theoretic approach

a) Model Assumptions and Valid Model
The first step in Information Theoretic approach is to make sure that the full (or global) model satisfies assumptions and is a good fit to the data.

In this case, I would suggest that the authors check for multicollinearity (ie. use a metric like VIF to identify problematically correlated variables). For example, time since last male visit and time since last visit with pedal marking might be problematically correlated.

Also check for overdispersion, and address it if necessary. With many zero visit days, overdispersion is a risk and makes the results look more 'significant' than they really are. Ben Bolker has a good reference for testing for, and addressing overdisperion in lmer (https://bbolker.github.io/mixedmodels-misc/glmmFAQ.html#overdispersion)

The full models should also be checked to make sure they fit the data reasonably well i.e., calculate the marginal and condition R2. Nakagawa and Schielzeth (2013) present a good way of doing so for mixed models and the MuMIn package includes a function based on their method: r.squaredGLMM().

Nakagawa S, Schielzeth H. 2013. A general and simple method for obtaining R2 from generalized linear mixed-effects models. Methods in Ecology and Evolution. 4(2):133–142.


b) P-Values
Information Theoretic (IT) approach is an approach for model selection. So technically if AIC is used to select a model, the traditional results of that model (including P-values) can be presented. However, most researchers using AIC do not present p-values because of the philosophical differences between IT and Hypothesis testing. In the IT framework, it is better to present confidence intervals (CI) around the effect. Since, mathematically CIs work out pretty much the same as P-values, cause less friction (philosophically), AND are generally considered superior to P-values (even in hypothesis testing), I suggest presenting the results as the effect with 95% Confidence Limits (see below for how to do this with MuMIn).


c) IT Model selection and interpretation
Using IT in this study is makes perfect sense, given the type of questions being asked. However, I feel that the process isn’t quite finished. The main problem is that, with one exception, every analysis ends up with multiple models, and it is unclear why only the ‘best’ of these multiple models is presented in Table 3 (one of the main benefits of IT is that is recognizes that there may be several models with similar support). If only these models are considered, the analysis looses a lot of information.

Using IT like this, to choose just the best model does happen, but not advisable (because of situations like this, where many of the top models are pretty close to each other).

I would recommend model averaging. The MuMIn package in R has a built in function for this: model.avg(). Instead of using a cutoff of 2 delta AIC, I would use the 95% confidence set (which will often include more models, but has better theoretical support than an arbitrary cutoff). It is straightforward to also calculate the confidence limits around the averaged parameters. There are specific examples in the documentation of MuMIn’s model.avg() function for how to achieve this in R (ie. ?model.avg).

This reference provides details on model averaging and on using the 95% confidence set:

Symonds MRE, Moussalli A. 2011. A brief guide to model selection, multimodel inference and model averaging in behavioural ecology using Akaike’s information criterion. Behavioral Ecology and Sociobiology. 65(1):13–21.

An overall article that addresses many of these issues (https://peerj.com/articles/4794/)

Harrison XA, Donaldson L, Correa-Cano ME, Evans J, Fisher DN, Goodwin CED, Robinson BS, Hodgson DJ, Inger R. 2018. A brief introduction to mixed effects modelling and multi-model inference in ecology. PeerJ. 6:e4794. doi:10.7717/peerj.4794.


d) Presentation of the results
Right now these results are a bit confusing because the results overlap between Table 3 and Table S7 and could be clarified.

I would recommend making Table 3 the results of model averaging including effects +/- confidence intervals, full model R2, etc. Include all parameters in the confidence set of models, whether or not the CI 95% includes 0. This makes it easy for readers to quickly scan the results and see which parameters are interpretable and which are not. Also be sure to clarify what each parameter means in the table (i.e. what does a positive effect of season mean?). Note that here, the effect of age_sex is always compared to females, i.e. each estimate shows the difference between an age-sex class and the females. As males seem to be the most different, perhaps they should be used as the contrast (or consider a post-hoc test to tease apart the differences).

In each analysis, there are not too many models in the candidate set. Therefore, I would suggest including all models for each analysis in the Table S7, ranked by AICc (as they currently are), and using cell shading or some other type of formatting to indicate of which models were in the confidence set (i.e. which included in model averaging) and which were not. I would remove all parameter estimates from Table S7, and simply indicate whether or not a parameter was included in the model or not.

An example of the presentation of results, both model averaged results (Tables 2 and 3) and model selection (Appendix tables). Note that Tables 2 and 3 present the results of three different models each (one in each column). This is just an example, there are other ways of presenting the results.

LaZerte SE, Otter KA, Slabbekoorn H. 2015. Relative effects of ambient noise and habitat openness on signal transfer for chickadee vocalizations in rural and urban green-spaces. Bioacoustics. 24(3):233–252. doi:10.1080/09524622.2015.1060531.

---

## Round 0.3 · Minor Revisions

I am sorry to hear that you had a tick-borne illness and glad that you have recovered. This risk is certainly something that has concerned me over the years.

Reviewer 2 of the previous version was not available to examine the revision. Reviewer 1 recommended rejection, indicating that he/she did not feel that the manuscript was yet ready for publication but that the authors were unwilling to make appropriate changes and were resistant and rude to the reviewers. Reviewer 3, who was invited for her statistical expertise, on the other hand, indicated that most of her suggestions that were not followed were not a problem.

I did not find the rebuttal comments to be offensive, although there might have a somewhat more abrupt approach to the reviewers than to the editor and I did find that the changes were not made as thoroughly as I would have hoped (details below). It is important not to be too quick to accuse reviewers of failure to understand. When a reviewer fails to understand, there is frequently (but not always) a reason in the writing. When writing about your own study, things may be so obvious that it is not obvious to a new reader. Sometimes an author assumes failure to understand but has himself misunderstood the reviewer’s point, perhaps because the reviewer expressed it poorly or incompletely. [For example, in your rebuttal document: “Editor: L202. On the pdf, I suggested a rephrasing to conform to the fact that this is a correlational not experimental study.” (This referred to as statement from the previous version: “we analysed which descriptors could have an effect on the observed patterns.), Your Reply: “In no place we say (or said) it is an experimental study. Line 202 starts with “we described the overall use of the site and the behaviors...” meaning that the analyses are descriptive and not aimed at making predictions (which is a further type of use of analyses). In my opinion it is much better saying what the study is and not what it is not.” What was missed here was that I changed the words ‘have an effect’ which implies causation and therefore require an experimental test with ‘were associated with’ which implies correlation as appropriate for your descriptive study.] In general, the authors bear the primary responsibility for the validity of the article, and I am willing to allow them their preference as long as it does not compromise the science or the clarity of the message.

I would like this revision to be the last that is required. If there are any uncertainties about what I mean or how you should respond, you can contact me directly at donald.kramer@mcgill.ca so that we can sort them out before resubmission.

I agree with all but two suggestions of Reviewer 3, Stephanie LaZerte:
• Although I would not object to adding the suggested sentence to the conclusions in the Abstract, I think it would be somewhat redundant and am comfortable with the Abstract as it stands.
• With regard to p-values, I understand the arbitrariness of 0.05 as the alpha level. Nevertheless, I do agree that one should not present explanatory variables with p-values of 0.06 as significant without explanation. However, I strongly prefer that you do not change your alpha level. There is a solid precedent for referring to cases where p is greater than 0.05 but less than 0.1 as ‘tendencies’. I suggested such wording on the two cases that I noticed this (L263, 286).

Some issues raised in the rebuttal document require additional comments.

You argued that my corrections of text were a matter of personal preference and that authors should be allowed some leeway. I agree that authors should be allowed some leeway as long as they are using correct English word choice and grammar. Looking again at my comments on the previous version, I think most of my suggestions do involve correction of grammatical and wording errors. However, I found that you made appropriate changes for the majority.

In one case, highlighted in your rebuttal, where you did not change, the reason for changing ‘descriptors’ to ‘variables’ was that you had not previously used the term ‘descriptors’ and had no reference to Table 1 where that term is used. I think that the term descriptors referring to behavior would not be immediately clear to most readers. A descriptor of marking behavior might be its intensity or duration, for example. On rereading, I see that the follow sentence does explain the potential descriptors. However, this will not be clear to readers because there is no link such as ‘The selection of descriptors was based on our hypotheses that . . .’ I suggest citing Table 1 at the end of the sentence where you mention descriptors in L200 and changing the term to explanatory variables because most readers would not perceive time since previous visits or rainfall to be a descriptor of behavior. (This will also require a change in the sub-heads of Table 1.)

Looking into this issue also made me realize that you had not mentioned the source of the weather data including how far the weather station was from the study site. I suggest adding this information at the end of the Sampling Protocol section.

You indicated that you found my suggestion that the units in Fig. 2 should be the number of behavioral acts per visit ‘confusing’. I will try to explain my reasoning again using an example. In Fig. 2B, females contribute about half as many acts of sniffing the tree as males do. How much of this difference is due to the lower frequency of female visits and how much to a lower probability of sniffing on a given visit? It makes more sense to me and seems more interesting from a behavioral ecological perspective to separate these two components by providing the proportion of visits by each sex and age class (which you do in the text) and separately the proportion of visits by each class in which tree sniffing occurred (which you could do in Fig. 2). Despite feeling that this would be clearer and more useful, I will leave it up to you to decide whether to change.

You indicated that you had checked the manuscript for consistent spelling convention. However, the text now uses U.S. spelling, but the tables and their headings use U.K. spelling for behaviour. (Please check that supplementary tables and figures are also consistent.)

Although you state that you tried to clarify when you were referring to brown bears or bears in general, your changes are inconsistent as the reviewer pointed out. L97 states that you will use the term ‘bears’ for ‘brown bears’ but you refer to brown bears even in the same paragraph as well as later ones, including statements related to this particular study. I cannot detect a consistent pattern.

Although you object to the reviewer’s concerns about this being a long-term study and state that you did not use this term, this is not correct. You did refer to a long-term dataset (L103 in version 1) which is conceptually the same as a long-term study. ‘Long-term’ in the present manuscript seems to apply only to the marking site where it is appropriate.

Additional editor comments

I have provide an annotated pdf to suggest small corrections.

L148. There is confusion about what constitutes a visit.
• L148 defines visit-events but not visits, so it seems that visit-events are the units of study and that these visit-events may be what is referred to in the text and Table 1 as visits. However, L225 refers to both visits and visit-events, with fewer visits, presumably because a bear’s visit could include multiple visit-events. L228 refers to the number of visits by males, but Table 2 shows that this number is a component of the number of visit-events, not visits.
• Fig. S1 shows the cumulative percent of videos with increasing intervals between them. However, the text indicates that there were 329 videos (L224-225) and the graph continues to 360 videos. Furthermore, a cumulative percent should reach 100% when all the gaps are considered, but the maximum appears to be at 40%.
• It is unclear how the analysis of gaps between videos can be used to define the duration of a visit-event. It seems to me that Fig. S1 can be used to support a criterion of a 20-min interval (e.g., ‘Videos separated by less than 20 min were considered part of the same visit-event’) but not a criterion of a 20-min duration (‘Videos occurring within 20 min of the first evidence of bear presence were considered part of the same visit-event’). There are situations in which a researh trying to repeat you study would be unable to decide what to do. If a series of videos lasted longer than 20 min but contained no interval between videos longer than 20 min, was it considered to be a new visit-event or was it not counted at all because it was more than 20 min after the first evidence of a bear present? What does ‘first evidence’ mean – on that day or after some interval? If a bear appeared two hours later, is it not considered a visit-event at all because it comes more than 20 min after first evidence?
• It is very important to clarify a) what you mean by all the terms you use, b) precisely how they were calculated so that another researcher could repeat them reliably, c) which variable was used in analyses (visits are referred to multiple times in Table 1) and d) to be sure that you use the terms consistently throughout the text, tables and supplementary material.

L195. Since your analysis considers the number of behavioral acts (L247ff), it is also important to define what constitutes a behavior. Can more than one act of the same type be counted in a visit-event? If so, what temporal separation is required for two instances of the same behavior within an event? If not, specify that your data consisted of whether or not each behavioral category occurred one or more times in each visit-event.

L231. The statement ‘visits follow the typical bimodal diel pattern’ has several problems. First, If you are referring to activity patterns specifically, not other diel patterns, you should be specific. Second, I think you are referring to dawn and dusk, not specifically only during sunrise and sunset. Third, you need a reference so readers can see what the typical bimodal activity pattern is. Fourth, activity patterns are ‘typical’ vary between species and sometimes for populations within species; if you are referring to brown bears, you should be specific. However, in reviewing the literature for a project of my own, I found lots of variation in activity patterns of bears (although I don’t recall how much applied specifically to brown bears). Thus, a clear reference is needed. Finally, and most importantly, I don’t think the figure provides very strong support at all for an evening peak. By combining data for most months, you may have obscured any pattern because of the changing time of dawn and dusk. In addition, you have not specified what time zone the x-axis indicates and whether you corrected for daylight savings time. I know it is a minor point for the paper, but if you want to draw a conclusion, that conclusion needs to be supported by the data. By stating that bears are active throughout the day (L232), there is an implication that they are not active at night. Fig. S3 does not support this implication. Furthermore, you should specify ‘visits’ not activity, because you were not measuring activity in general and this is a case where there could easily be a difference between general activity and communication.

L261. I don’t see support for a seasonal effect in Table 2. You probably mean Table 3.

Fig. 2. Switch panels so that the seasonal patterns are A and B because you refer to them first in the text.

Supplemental material
Fig. S1. This figure is missing a caption. Capitalize first letter in axis labels. y-axis: Cumulative proportion of videos.
Fig. S2. Missing caption. Needs scientific names. Needs identification of lines showing variability. Capitalize total on right y-axis.
Fig. S3. Missing caption. Use black, not gray. Remove horizontal lines. Capitalize Number.
Supplementary tables are not in order cited in text.
Table S1. undetermined (correct spelling)
Table S2. Capitalize Wild boar (consistency). Add scientific names. Add a column for camera days.

Reviewer 1 ·

Basic reporting

The grammar could be better, and some of the terminology is confusing. For example, the authors say 'brown bears (hereafter bears)', but then continue to switch between 'brown bears' and 'bears' for the rest of the paper.

Previous reviewers previously made suggestions on how to improve the readability of the manuscript, but the authors rebutted most of the suggestions. The comments were not personal attacks on the authors, but were apparently taken as so, leading the authors to take a combative stance. Even if you do not like the proposed changes, they point to areas of the manuscript that need improvement. Just rebutting suggestions based on "personal preference" doesn't fix the problem.

The authors are clearly uninterested in further peer review of the manuscript, but I don't believe it is ready for publication. I would encourage more mature responses in the future.

Experimental design

Descriptive.
See past reviews. Authors do not seem interested in considering other methods of analyses. They also refer to past versions of the manuscript, which readers will not have access to, when rebutting changes.

Validity of the findings

Valid.
Limited by being just one site.
Limited by analyses used (and authors are unwilling to consider others).

Additional comments

I would encourage the authors to make more of a good faith effort to address reviewer comments in the future. I can understand the authors may be frustrated, but being rude to reviewers doesn't do anyone any good. This was possibly the worst response to reviewer comments I have ever seen.

·

Basic reporting

Including conclusions in the abstract

The authors do include detailed results in the abstract, and broad conclusions, but not what I would consider a study-level conclusion. For example, in the discussion on L310-312, a clear conclusion is made: “Our results, in accordance with published information, show that tree rubbing can be performed by any class of individual at any time, but it is clearly monopolized by adult males, especially during the mating season”. I would suggest that adding this kind of statement to the abstract makes the abstract much more interesting and informative. However, I feel a bit like I’m quibbling over the details, so I leave it up to the authors to decide whether or not they want to make any changes.


L223. Consider adding a small note explaining that pedal marking models only included males because it was performed almost exclusively by males. I realize this information is in the results, in Table 1, AND in Table 3, but there are many models and a lot of information which can make it easy for a reader to miss important details. As this is the only model that deals just with males, adding something here might help guide the reader along.

Experimental design

Alpha
In the results, authors consider all effects where the P < 0.10. This is completely fine, but most readers would assume an alpha of 5%. Therefore, I suggest noting in the methods that an alpha of 10% was used to evaluate the results.


L218 R packages should get their own citations both in text and in the references. This ensures the authors of open source software get credit for their work. Including the package version makes the research more reproducible. You can use the R funciton citation(“gml”) to get the recommended citation for any package, as well as the function packageVersion(“gml”) to quickly get the version of the package.

For example (using citation info for lme4 and MuMIn, and the versions on my computer):

Analyses were performed in R v3.3.3 with packages gml(vX.X.X REF), lme4 (v1.1.21 Bates et al. 2015), and MuMIn (v1.43.6 Bartoń 2019).

Kamil Bartoń (2019). MuMIn: Multi-Model Inference. https://CRAN.R-project.org/package=MuMIn

Douglas Bates, Martin Maechler, Ben Bolker, Steve Walker (2015). Fitting Linear Mixed-Effects Models Using lme4. Journal of Statistical Software, 67(1), 1-48. doi:10.18637/jss.v067.i01.

Validity of the findings

1. Pseudoreplication
I like this compromised approach. When we don’t know the unknown individuals, there is the worry that it is mostly one unknown individual. In that case, controlling for it in the model would be the correct approach. If, on the other hand, the unknowns are a large mix of different individuals (as the authors suggest), it won’t matter (as the authors found). Any readers who wonder can take a look at the supplemental material.

2. P-values
I’m fine with p-values if the authors feel strongly about them. There is definitely something to be said for including them as a familiar statistic.

3. Model Selection
The authors present compelling references for not doing model selection, especially with this dataset (observational with the potential for some correlated variables, even if multicollinearity is addressed).

I am satisfied with this rational, and have only the suggestion that they add a sentence describing how they select which models to interpret, something like:
L216/217 “From the resulting models, we report only those within ΔAIC<2, and we interpret only the best model (that with the lowest AIC)”
OR
“From the resulting models, we report and interpret only those within ΔAIC<2”.
Which ever best describes their approach.

4. Presentation of the results.
I’m glad the authors included R2 and am fine if the authors feel like they would prefer to keep the current formatting.

However, I still feel that it should be clear in Table 3 that in tree rubbing, the effects of age_sex are all compared to the base group, females. For example, in the text (L282-284), it states that “Adult males and juveniles had higher probabilities of tree-rubbing during their visits than the rest of the individuals”.

This isn’t correct: adult males and juveniles both perform more tree rubbing than females, but indeterminate individuals were not significantly different from females. While the estimates do suggest a significant difference between indeterminate and males/juveniles, there is no statistical test confirming these differences (a post hoc test would be needed).

I would suggest changing the statement on L282-284 to something like “Adult males and juveniles had higher probabilities of tree-rubbing during their visits than females.” And would also suggest altering the text in Table 3 to something like: “Undetermined vs. Females”, “Juvenile vs. Females”, and “Males vs. Females”. Something at least to explain to the reader why females don’t appear in the Table 3.

Minor Comment:

L358-368. The “sniff tree” model only has an R2 of 0.06, which is low even for ecological/behavioural studies. Consider adding a note to that effect.

Additional comments

Overall I am satisfied with the authors responses to my comments. I have a few outstanding issues/suggestions which I have highlighted, but I do not think they are extremely problematic. I note again, that I have mostly focused on methods, and particularly on the statistics.

---

## Round 0.4 · accepted · Accept

The authors have responded adequately to the suggestion of the reviewers and editor. I now consider it ready for publication.